# Symmetry breaking propulsion of magnetic microspheres in nonlinearly viscoelastic fluids

Louis William Rogowski[1], Jamel Ali[2], Xiao Zhang[1], James N. Wilking[3], Henry C. Fu[4✉] & Min Jun Kim [1✉]

Microscale propulsion impacts a diverse array of fields ranging from biology and ecology to health applications, such as infection, fertility, drug delivery, and microsurgery. However, propulsion in such viscous drag-dominated fluid environments is highly constrained, with time-reversal and geometric symmetries ruling out entire classes of propulsion. Here, we report the spontaneous symmetry-breaking propulsion of rotating spherical microparticles within non-Newtonian fluids. While symmetry analysis suggests that propulsion is not possible along the fore-aft directions, we demonstrate the existence of two equal and opposite propulsion states along the sphere's rotation axis. We propose and experimentally corroborate a propulsion mechanism for these spherical microparticles, the simplest microswimmers to date, arising from nonlinear viscoelastic effects in rotating flows similar to the rod-climbing effect. Similar possibilities of spontaneous symmetry-breaking could be used to circumvent other restrictions on propulsion, revising notions of microrobotic design and control, drug delivery, microscale pumping, and locomotion of microorganisms.

---

[1] Department of Mechanical Engineering, Southern Methodist University, Dallas, TX 75205, USA. [2] Department of Chemical and Biomedical Engineering, FAMU-FSU College of Engineering, Tallahassee, FL 32310, USA. [3] Department of Chemical and Biological Engineering, Montana State University, Bozeman, MT 59717, USA. [4] Department of Mechanical Engineering, The University of Utah, Salt Lake City, UT 84112, USA. ✉email: henry.fu@utah.edu; mjkim@lyle.smu.edu

Propulsion at the microscale is limited by physical constraints that can be analyzed using symmetries. At such small scales, the hydrodynamics are described by the viscosity-dominated Stokes equations[1], and propulsion mechanisms are often limited by their geometric symmetries and the physical laws governing Stokesian hydrodynamics. For example, in Newtonian fluids, force- and torque-free biological swimmers have been analyzed in terms of time-reversal symmetry of Stokes flow, with the resulting Scallop theorem[2] requiring that kinematic reversibility be broken by non-reciprocal strokes to achieve propulsion[3,4]. The same principle has been instrumental in the development of microrobotic propulsion, where kinematic reversibility must be broken either through non-reciprocal flexible response to reciprocal actuation, or by non-reciprocal actuation, such as steady rotation from magnetic fields—the type we focus on in the remainder of this paper. For these, the analysis of hydrodynamic mobility matrices reveals that geometries more symmetric than a flat triangle do not couple torque to linear translation[5,6] and hence cannot be propelled by rotation; furthermore, symmetry analyses of the entire dynamical system, including actuating fields and moments, have shown that there is no average propulsion of rotated microrobot populations that are symmetric under combined charge conjugation and parity[7].

In non-Newtonian fluids, time-reversal symmetry is explicitly broken and kinematic reversibility does not hold, allowing reciprocal strokes to achieve propulsion[8–10]; however, geometrical symmetry analyses still apply. Of particular importance here is fore-and-aft symmetry relative to the direction of propulsion. It is usually thought that propulsion is not possible in a fore–aft symmetric system; for if there is a single state with propulsive velocity, upon fore–aft reflection the velocity ($U$, Fig. 1a) must be simultaneously reversed (to $U'$, Fig. 1a) and equal to the original, and hence be zero[11]. For example, experiments achieving propulsion by reciprocal actuation of dumbbells in nonlinearly viscoelastic fluids[12] instead explicitly break fore–aft symmetry by employing either dumbbells with two different sized beads, or a boundary nearby symmetric dumbbells. In a theoretical analysis of axisymmetric swimmers rotated along their axis of symmetry in nonlinearly viscoelastic fluids, fore–aft symmetry led to the conclusion that the simplest shapes capable of propulsion are snowmen constructed of two differently sized spheres[13]. The same symmetry considerations suggest that it is not possible for a steadily rotating sphere to be propelled along its rotation axis.

However, this symmetry analysis neglects the possibility of spontaneous symmetry breaking, in which a pair of translating states exists with equal and opposite propulsion velocities, rather than a single state with zero velocity. Here, we report the spontaneous symmetry-breaking propulsion of rotating spherical magnetic microparticles within two different non-Newtonian fluids: a low concentration polyacrylamide solution and a synthetic mucus solution. Application of a static magnetic field can select between the two states, enabling control of the propulsion direction. We propose a physical mechanism for symmetry breaking that arises from nonlinear viscoelastic effects in rotating flows, similar to the rod-climbing effect, which pushes the sphere along its rotation axis. The mechanism is corroborated by comparison to existing theoretical analyses of rotating and translating spheres in generic third-order fluids[14]. Thus, we demonstrate that spontaneous symmetry breaking can be used to propel and control rigid spherical magnetic microparticles without requiring magnetic gradients, geometry-altering surface coatings, or catalytic propulsion. Our results will enable improved applications of microparticles, especially biomedical, as well as insights into locomotion of living and artificial microswimmers in complex fluids.

## Results

**Microparticle propulsion and directional control**. Spherical magnetic microparticles were observed to translate while exposed to uniform rotating magnetic fields in both a 4% mucin solution and a 0.25% polyacrylamide solution. Motions both perpendicular (transverse) to and along the field's rotation axis were observed within both non-Newtonian fluids. We attribute the transverse motion to the well-understood rolling motion along the boundary caused by the particles' rotation[15,16], since the transverse velocity weakened as the distance from the boundary increased, reversed upon reversal of the rotation direction, and had a linear relationship with frequency nearby the boundary (Supplementary Figs. 1–3). On the other hand, the motion along the rotation axis was surprising since it is symmetry-prohibited (Fig. 1a). In the rest of this paper we focus on this symmetry-prohibited motion, hereafter referred to as propulsion.

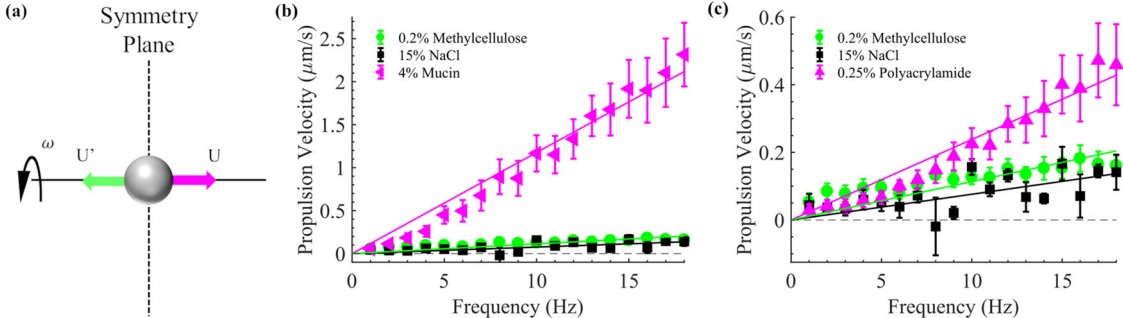

**Fig. 1 Propulsion characteristics of microparticles. a** Reflection about the symmetry plane leaves the geometry and rotation unchanged but reverses the propulsion velocity $U$ to $U'$. If there is only one state, $U = U' = 0$, the spontaneous symmetry-breaking results in two propulsive states with equal and opposite velocities $U$ and $U'$. Propulsion velocity vs. rotational magnetic field frequency for 10 μm diameter microparticles in (**b**) 4% mucin and (**c**) 0.25% polyacrylamide. Control experiments using 10 μm diameter microparticles were performed in two Newtonian fluids: 15% NaCl and 0.2% methylcellulose; both were plotted in (**b**) and (**c**) to compare with the propelling microparticles. The coefficient of determination ($r^2$) values for the linear fit in 4% mucin solution and 0.25% polyacrylamide solution were 0.9650 and 0.9449, respectively, indicating a strong linear correlation of velocity and frequency. In contrast, the $r^2$ values for the linear fits to controls in 15% NaCl and 0.2% Methylcellulose solution were 0.3773 and 0.2036, respectively, indicating little correlation with frequency. Data are presented as mean ± standard error. The number of independent particles averaged together was six for 4% mucin, seven for 0.25% polyacrylamide, nine for 0.2% methylcellulose, and four for 15% NaCl; each individual particle had at least three independent trials. Source data are provided as a source data file.

All experiments reported were conducted far from the substrate surface (>100 μm) to minimize boundary effects (Supplementary Fig. 1) unless otherwise stated.

When the magnetic field rotated in a plane, the propulsion direction was seemingly randomly oriented either along or opposite the rotation vector (perpendicular to the plane) and independent of the rotation direction. However, we found that imposing a symmetry-breaking static magnetic field (2 mT) along the rotation axis fixed the direction of propulsion and we used this to investigate the dependence of propulsion velocity on rotational frequency (Fig. 1b, c). Both 4% mucin and 0.25% polyacrylamide are viscoelastic and shear thinning at the applied shear rates[10,17–19] and possessed first and second normal stress differences (see in Methods). Polyacrylamide solutions are known to have first and second normal stress differences[20,21], and we were able to measure normal stress differences in 10% mucin solutions (see in Methods and Supplementary Note 10). In both fluids there was a nearly linear relationship between propulsion velocity and frequency. Note that the propulsion decreased to almost zero as the rotational frequency approached zero, implying that static magnetic field gradients are not responsible for propulsion; see Supplementary Note 8 for further experiments that rule out this possibility. In control experiments using the Newtonian fluids 0.2% methylcellulose[22] and 15% NaCl, we observed small, nearly vanishing, propulsion velocities that were not correlated with frequency. Since this small propulsion velocity was observed even without any actuation (at near-zero frequency), we attribute it to an internal flow within the sample chambers. Propulsion was ubiquitous among rotated microparticles; it was observed for beads of several different diameters (2, 4, 8 μm) in mucin solutions of various concentrations (2%, 3%, 4%) and in the polyacrylamide solution (see Supplementary Note 9). Propulsion was observed for >90% of beads in mucin solution, and >60% of beads in polyacrylamide solution.

The observed propulsion was repeatable and could be controlled under both open-loop and closed-loop feedback control schemes (see Methods for details). Figure 2a shows an open-loop trajectory in which the microparticle was directed to create a square shape. Transverse rolling slightly skewed the trajectory in Fig. 2a and was influenced by the microparticles' rotation direction (Supplementary Note 2). During these experiments, the microparticles sometimes encountered localized regions of hindered propulsion, as seen in Fig. 2b, where the microparticle could not perform the desired trajectory during the first downward segment; further examples can be seen in Supplementary Fig. 1b where propulsion velocity varies with location in the mucus medium. We suspect that these localized regions correspond to elevated concentrations of mucin glycoprotein entanglements as they were far less common in the homogeneous polyacrylamide solution. Given enough time, microparticles usually navigated around these regions and completed their intended trajectories. Closed-loop feedback control was used to navigate microparticles along trajectories, such as those in Fig. 2c–g, where the target points and desired trajectories are shown as magenta dots and dashed lines, respectively. Microparticles usually generated trajectories relative to the intended paths with small levels of error (Supplementary Fig. 8). Finally, open-loop control in 3D was achieved (Fig. 2h, details in Supplementary Note 4) during which the microparticle was directed to propel along the positive x-direction, along the negative z-direction (into the page), along the negative y-direction, along the positive z-direction (out of the page), and then finally along the positive x-direction again. Although simple, this trajectory shows that 3D control is feasible using these microparticles and trackable by correlating the pixel area of the microparticle to its z-position above or below the focal plane

(Supplementary Fig. 9a–c). Supplementary Note 4 details 3D propulsion of a microparticle within 0.25% polyacrylamide. Notably, since propulsion generally continued as microparticles translated through the fluids, rather than being limited to specific regions, these demonstrations of control rule out interactions with specific heterogeneities as the origin of the propulsion[23].

**Static magnetic field iteration and symmetry-broken states.** In the above experiments, application of a magnetic field controlled the propulsion to occur in the direction of the static magnetic field, so we next investigated the dependence of propulsion on the static magnetic field. We gradually changed the applied static magnetic field from −5 to 5 mT in 1 mT increments while rotational frequency remained constant, and observed the average propulsion velocity of 10 μm microparticles at each increment (Fig. 3a, details in Methods). As the static magnetic field approached 0 mT, the average propulsion velocity switched direction, from negative to positive, while remaining nearly constant for larger positive and negative fields. To further understand this phenomenon, the static magnetic field was varied at smaller 0.2 mT increments (Fig. 3b), revealing that the switching field strength depends on the increment size; for smaller increments the average propulsion direction did not switch until the static field reached 0.8 mT for this particular microparticle in 4% mucin. Furthermore, upon tracking the velocity of individual microparticles in 4% mucin (Fig. 3c), we found that each particle switches propulsion direction at drastically different static fields with only a slight variance of the switching field observed between individual trials (Supplementary Fig. 4a). When performing the same experiments in 0.25% polyacrylamide (15 Hz), the variation between individual microparticles was lost (Fig. 3d) and instead all microparticles switched directions close to 0 mT. Thus, we consistently observe the existence of propulsion and switching of propulsion direction as the static field is swept, although there is strong variability in the propulsion velocity and switching frequency for different static field histories and increment sizes (see Supplementary Note 2), which could be due to factors such as interparticle variation and medium heterogeneity (see Supplementary Note 9). In the remainder of the paper we focus on understanding the robust existence of propulsion and whether the static magnetic field is necessary for propulsion to occur.

To elaborate on how a static field might be able to control the propulsion direction of a rotating microparticle, first consider that in a Newtonian fluid, standard analytic solutions indicate that a sphere rotated by an external torque will not translate[24], consistent with the results in Fig. 1b, c. When the magnetic field rotates in a plane, the magnetic dipole of the sphere also rotates in the same plane and the sphere rotates along the same axis as the field. If there is also a static field along the rotation axis, the magnetic dipole tilts towards the static field and continues to rotate with the field, but now there is an additional rotation of the sphere along the axis of the magnetic dipole (see Supplementary Note 5). Despite the more complicated rotational dynamics, the microparticle in a Newtonian fluid will not translate. Remarkably, we have observed the consistent translation of such spheres in synthetic mucus and polyacrylamide. The more complex rotation and the static field explicitly break the fore–aft symmetry, but due to the robust observation of propulsion combined with variability in switching of propulsion direction, we hypothesized that there is an underlying spontaneous symmetry-breaking translation for a force-free sphere rotated by an external torque in a nonlinearly viscoelastic fluid. Such symmetry-breaking leads to pairs of translational states with equal and opposite velocities in the rotation direction, consistent with our observations above. The behavior observed with a symmetry-breaking static field can be thought of as selecting one symmetry-broken state over the other.

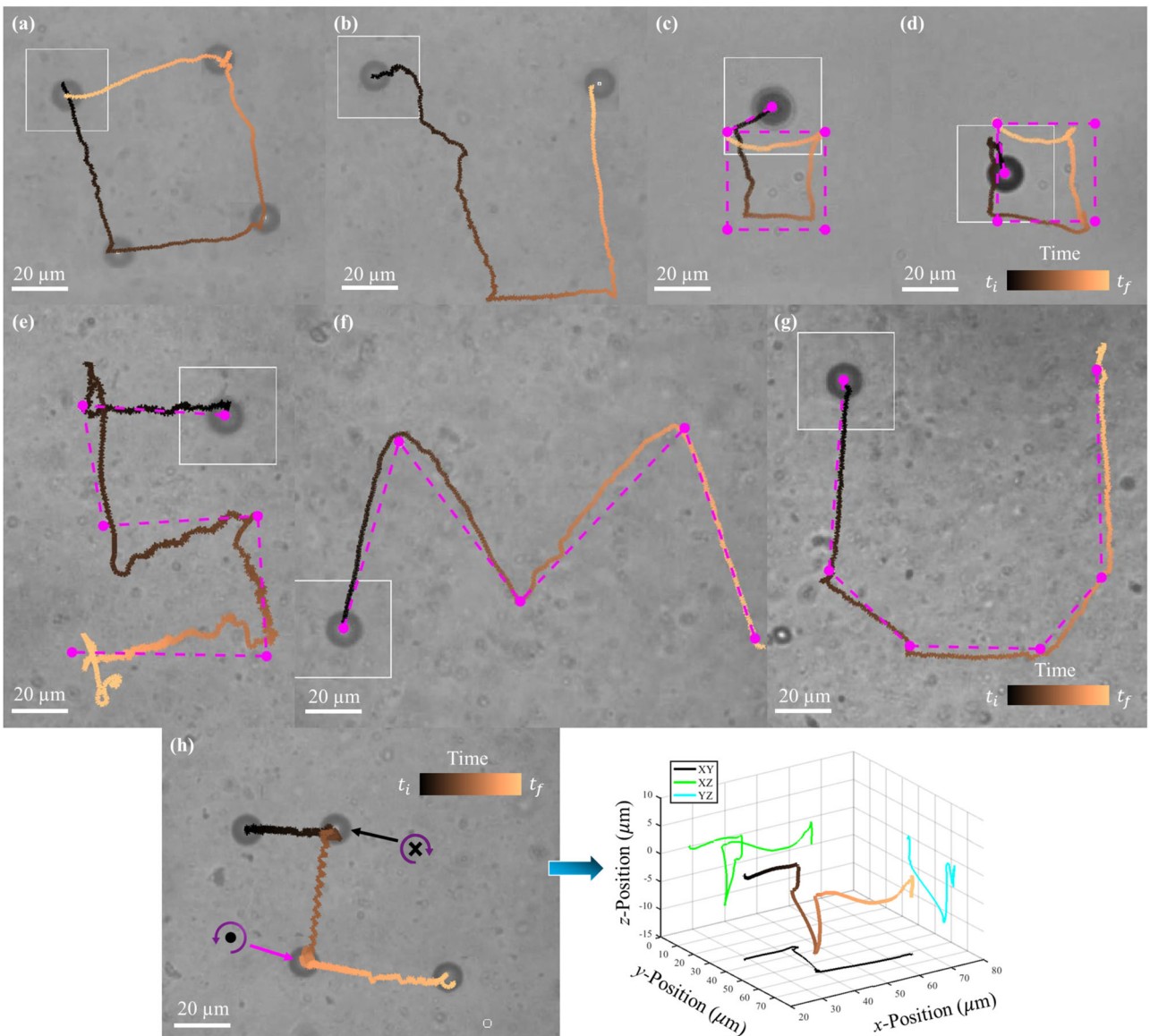

**Fig. 2 Path planning of microparticles in nonlinearly viscoelastic fluids. a** Open-loop control of a microparticle performing a square trajectory in 4% mucin. **b** Open-loop control of a microparticle in 4% mucin, which encounters a region of relatively high fluid resistance, disrupting its initial downward motion. Selected trajectories of a microparticle under closed-loop feedback control in 0.25% polyacrylamide (**c, d**) and 4% mucin (**e–g**), where magenta dots represent the intended targets and dashed lines represent shortest path. The times associated with each trajectory are (**a**) 80 s, (**b**) 95 s, (**c**) 156 s, (**d**) 145 s, (**e**) 94 s, (**f**) 60 s, and (**g**) 43 s. **h** Left: a simple 3D trajectory that was achieved using open-loop control. At the points indicated by the arrows, the microparticle translated downwards in the negative $z$-direction (black arrow) and then upwards in the positive $z$-direction (magenta arrow); the circular purple arrow signifies rotation direction from behind the heading vector, while the black circle and × represent out of the page and into the page, respectively. Right: the estimated 3D trajectory of the microparticles path and projections along the different planes; the total time of the trajectory was 114 s. Microparticles were rotated at (**a–c, e–g**) 19 Hz, (**d**) 40 Hz, and (**h**) 15 Hz. Source data are provided as a source data file. These experiments were repeatable for >90% of all particles examined in 4% mucin and >60% of all particles examined in 0.25% polyacrylamide.

To test if we could instead observe spontaneously (not explicitly) broken symmetry, we swept the static magnetic field from 2 to −2 mT and then back to 2 mT in 0.2 mT increments. As seen in Fig. 3e, the propulsion velocity in 4% mucin has a hysteretic response to the field strength, switching direction at around −1.2 mT (from positive to negative) and then switching back at around 0.8 mT (from negative to positive). The experiment in 0.25% polyacrylamide showed much sharper changes between propulsion states (Fig. 3f), but at 0 mT two separate propulsion states could still be achieved, although only at a higher rotational frequency (40 Hz). In the Supplementary information (SI; Supplementary Figs. 4b and 5) we show that the switching points

depend on magnetic field increment and frequency. Thus, in both fluids, at an applied static field of 0 mT, we could observe both directions of propulsion for both fluids (on the downward and upward sweeps), corresponding to truly spontaneously symmetry-broken propulsive states. We were consistently able to demonstrate symmetry-broken propulsive states in microparticles that showed propulsion. In mucin solution, all particles tested showed hysteresis when subjected to negative and positive sweeps of the magnetic field. In polyacrylamide solution, all 4 particles tested showed hysteresis (see Supplementary Note 2 and Supplementary Fig. 6), although we note that the strength of hysteretic effects was dependent on frequency; when actuated at

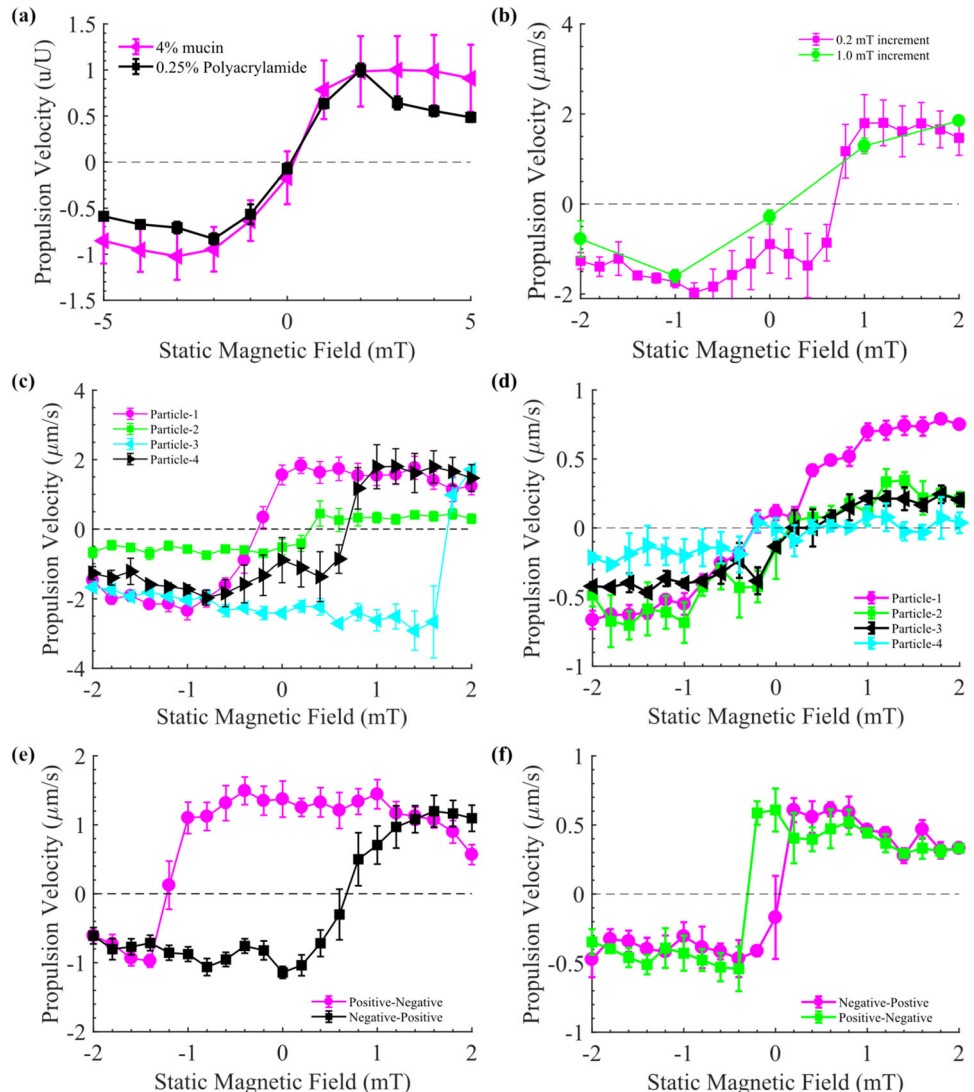

**Fig. 3 Effect of static magnetic field on propulsion.** Velocity vs. static magnetic field for 10 μm diameter microparticles for: **a** static field swept at 1 mT increments between the range of −5 to 5 mT, with the microparticles rotated at 15 Hz. Four microparticles were examined and each had at least three trials in each fluid. **b** Static magnetic field swept at 0.2 and 1 mT increments for the same microparticle in 4% mucin. **c, d** Static magnetic field swept at 0.2 mT increments for 4 different particles for 4% mucin and 0.25% polyacrylamide, respectively. **e, f** Propulsion hysteresis in response to static magnetic field sweep at 0.2 mT increments for a single microparticle in 4% mucin and 0.25% polyacrylamide, respectively. The frequency for (**f**) was 40 Hz with the rotating magnetic field amplitude set to $|B_r| = 0.1750f$, where $f$ is the frequency in Hz. In all scenarios, velocities of microparticles remained relatively constant at larger positive and negative fields, but the field at which velocity switched direction depended on static magnetic field history, increment size, and specific particle characteristics. Dashed lines in graphs represent zero propulsion velocity while lines between points in (**a–f**) were added to act as a guide to the eyes. Statistics: (**a**) Four particles were examined in 4% mucin and six particles in 0.25% polyacrylamide; each microparticle had at least three independent trials. (**b**) Five trials were conducted at 1 mT and four trials at 0.2 mT. (**c**) Six, four, five, and four trials were conducted for particles 1–4, respectively. (**d**) Four trials were conducted for each particle examined. (**e**) Ten independent trials were conducted for a single microparticle. (**f**) Four independent trials were conducted for a single particle. Data are presented as mean±standard error. Source data are provided as a source data file.

15 Hz, very little hysteresis was observed and microparticles switched propulsion direction at close to zero static field for both positive and negative sweeps.

The hysteretic behavior of propulsive velocity is reminiscent of those seen in other spontaneously symmetry-breaking phenomena, such as ferromagnetism, but ferromagnetic behavior cannot explain our observations, since the coercive fields of 68 mT required[25] to magnetize and demagnetize our beads are much larger than the magnitudes of combined rotational and static fields used in our experiments (3.11–15.87 mT). Instead, we propose that the symmetry-breaking mechanism is caused by nonlinear viscoelastic stresses that arise in rotating flows, similar to those responsible for the "rod-climbing" effect. Around a

vertical rotating rod in a polymeric fluid, nonlinear first normal stress differences cause excess circumferential hoop stresses along the circular streamlines, while second normal stress differences cause excess radial stress; both can squeeze the fluid upwards around the rod[21]. We note that both polyacrylamide and mucin solutions display first and second normal stress differences that lead to rod-climbing-like effects (see in Methods and Supplementary Note 10). For a rotating and translating sphere, fluid material also advects past the sphere, so that fluid elements at the back of the sphere have spent more time circling the sphere than those at the front; thus, the back of the sphere has larger nonlinear viscoelastic stresses than the front, producing a net squeezing force that propels the sphere forward. A force-free

symmetry-breaking translational state occurs when this propulsive force balances the drag force from translation.

**Theoretical and experimental propulsion analysis.** To theoretically test this idea, we analyze the force on a sphere rotating with angular frequency $\Omega$ and translating with velocity $U$ using a retarded motion expansion in a generic third-order non-Newtonian fluid[21] (detailed in Supplementary Note 6). The force $F$ along the rotation axis was calculated as a perturbation expansion in the Deborah number $\mathrm{De} = \lambda\Omega$, where $\lambda$ is a timescale for fluid relaxation,

$$F = F^{(0)} + \mathrm{De}F^{(1)} + \mathrm{De}^2 F^{(2)} + \dots \qquad (1)$$

The zeroth-order velocity field is that of a rotating and translating sphere in a Newtonian fluid, and the zeroth-order force $F^{(0)}$ is given by Stokes' law for the drag of a translating sphere, $F^{(0)} = -6\pi\mu a U$, where $a$ is the microparticle diameter and $\mu$ is the fluid viscosity. The first correction to the force arises at order $\mathrm{De}^2$, and contains terms proportional to $\Omega^2 U$ and to $U^3$. The former term, $F^{(2)}_{\mathrm{thrust}} = C\Omega^2 U$, where $C$ is a constant that depends on the parameters of the constitutive law, corresponds to the symmetry-breaking propulsive thrust since it couples rotation and translation in a way that is independent of rotation direction[5]. If $C > 0$, then there is a nonlinear thrust in the same direction as translation which could stabilize a translating symmetry-broken state, i.e., a rotating non-translating sphere could become unstable to translation; while if $C < 0$, then nonlinear effects stabilize the zero-velocity state. Due to the instability of the non-translating state, upon rotating an initially stationary sphere any small perturbation would lead to the symmetry-broken translating state. The sign of the force contribution depends on the parameters specifying the third-order fluid, which are known for common fluid models. As detailed in Supplementary Note 6, we have found that the force contribution is positive for many model constitutive laws, implying that nonlinear fluid response can lead to symmetry-breaking propulsion in many types of non-Newtonian fluids. The form of $F^{(2)}_{\mathrm{thrust}}$ suggests that there may be a critical rotation rate at which it balances the drag $F^{(0)}$, and symmetry-breaking propulsion becomes possible. Finally, we note that although in a second-order fluid, rod climbing is directly related to the first and second normal stress coefficients that specify the constitutive law[21], our propulsive force is a higher-order effect that can only be consistently described using all the parameters that specify a third-order fluid, implying that nonlinearities beyond the normal stress differences impact symmetry-breaking propulsion.

To support this proposed symmetry-breaking mechanism, we used microscale particle image velocimetry (μPIV) to visualize the flow fields of microparticles within 4% mucin. These experiments were performed using a different experimental setup, but with operating parameters identical with the previous experiments (see Methods). Figure 4a–i shows instantaneous snapshots of the μPIV velocity flow fields in the $x$–$y$ plane and their indicated component velocities for: (4a–c) a microparticle that could not propel, (4d–f) a microparticle propelling, and (4g–i) a microparticle propelling along the positive $z$-direction. In Fig. 4a the propulsion velocity flow fields to the left and right of the microparticle cancel each other out, while the transverse velocities in (4b) are unidirectional; this results in (4c) having no net propulsion along the propulsion axis. In Fig. 4d the velocity flow fields do not cancel each other out, and the transverse velocity in (4e) to the left of the microparticle is no longer unidirectional; this results in (4f) having a net propulsion velocity. When examining a microparticle propelling in the positive $z$-direction, we found that in addition to a persistent azimuthal component (Fig. 4h), there is a radial component (Fig. 4g) that converges

towards the microparticle, resulting in the velocity flow field shown in Fig. 4i. Notably, in contrast to the flow around a rotating sphere in a Newtonian fluid which is purely azimuthal[24], the significant inward radial flow is consistent with the secondary flows expected from our hypothesized propulsion mechanism involving rod-climbing-like effects. Supporting our proposed flow mechanism, we found that the observed radial flow closely agreed with the secondary flow predicted by the theoretical perturbation expansion described above (see Supplementary Note 11 for details). Similar μPIV experiments for 0.25% polyacrylamide were also performed and can be found in Supplementary Note 12.

## Discussion

In this manuscript we focused on the existence and mechanism for symmetry-breaking propulsion. This type of propulsion may be ubiquitous in nonlinear fluids; we demonstrated it in two different fluids that have normal stress differences which can lead to rod-climbing-like effects. We have also observed that microswimmers composed of two beads with similar diameters can propel through mucus solutions[9], thus we expect many other symmetric geometries to behave similarly. However, further work is needed to completely understand the conditions (geometry, rotation frequency, fluid properties) required for symmetry breaking to occur, the size dependence and effect of boundaries on propulsion velocities, as well as how fluid properties and actuation history control the hysteresis and selection of propulsion direction. In addition, while the static magnetic field selects the propulsion direction, the precise mechanism by which the accompanying dipole tilting and dynamic rotation leads to selection remains poorly understood. The propulsive force we identified here may be related to recently observed reductions in drag on sedimenting and rotating spheres in nonlinear viscoelastic fluids[26] and the mechanism we described has some similarities to elastic instabilities in Couette-Taylor flows[27]; the analogy might shed light on criteria for the instability of the zero-propulsion state of our microparticles to symmetry-breaking states. Our perturbative analysis cannot find the symmetry-broken state, since that would require the second-order nonlinear contribution to cancel the zeroth-order drag, which violates the assumptions of the perturbative expansion; additional theoretical work that explicitly identifies symmetry-broken states in specific nonlinear fluids could also shed light on these questions, but would likely require direct numerical simulation.

For applications and broader impacts, the microparticles demonstrated here provide a direct avenue to improve existing medical applications. Spherical microbeads—easily fabricated and functionalized[28–36]—are already being utilized for tasks including hyperthermia[37,38], delivery of therapeutics[39], magnetic resonance imaging[40,41], and the formation of efficient microswimmers[42]. As mucus is ubiquitous throughout the human body[17] and hinders the transport of medically loaded micro- and nanoparticles[17,43–47], the controllable propulsion of spherical microparticles through mucus solutions opens the door to enhanced efficacy in biomedical applications. The mechanism of nonlinear symmetry breaking may provide insight into microscale pumping and the locomotion of microorganisms, which, along with artificial microswimmers, experience altered and sometimes improved motility in complex fluids[48–53]. Hoop stresses such as the ones we consider have been suggested to stabilize the trajectories of rotating bacteria[54]. In general, our work demonstrates that nonlinearities that do not explicitly break a symmetry may nonetheless enable microscale propulsion via symmetry breaking. Thus, it may be profitable to consider propulsion in other systems where it is seemingly disallowed by symmetry. Such symmetry-breaking propulsion has been predicted in autophoretic propulsion of chemically reactive

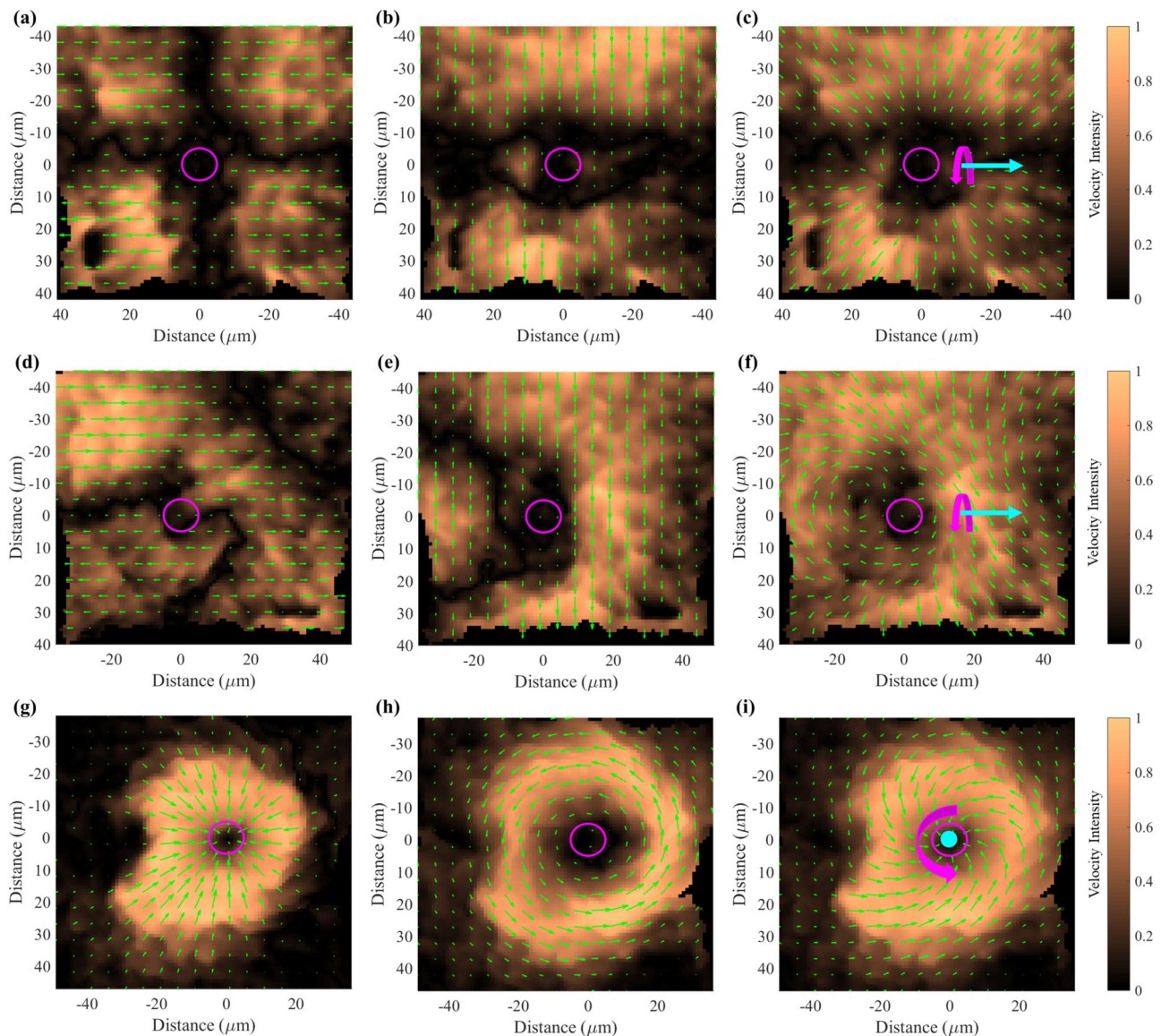

**Fig. 4 μPIV in *x–y* plane of microparticles in 4% mucin.** The (**a**) propulsion direction, (**b**) transverse direction, and (**c**) total velocity flow field for a microparticle not propelling. The (**d**) propulsion direction, (**e**) transverse direction, and (**f**) total velocity flow field for a microparticle that is propelling. The (**g**) radial, (**h**) azimuthal, and (**i**) total velocity flow field for a microparticle propelling away from the substrate (positive *z*-direction). Inward radial flow is consistent with secondary flows generated by hoop stresses. All μPIV experiments were performed close to the boundary (<100 μm, see Methods). Cyan arrows represent propulsion direction and magenta arrow represents rotation direction. Magenta circle is the microparticle. Color bars to the right represents percentage of velocity magnitude with respect to the maximum velocity magnitude. These experiments were repeated at least two other times and all results revealed similar profiles; the results in (**g–i**) were repeated in 0.25% polyacrylamide and can be found in Supplementary Note 12.

particles[55], and in purely mechanical systems outside the zero-Reynolds number limit, for heaving flat plates via inertial effects[56]. In microscale systems, we speculate that there may be other nonlinearities that may lead to spontaneously symmetry-breaking propulsion not only for artificial microrobots but also for living microorganisms.

Microrobots with geometry and actuation that obey fore–aft symmetry are often thought to be incapable of propulsion, but we have demonstrated that a spherical magnetic microparticle steadily rotated by a magnetic field spontaneously breaks symmetry to develop a pair of states with propulsion in opposite directions along the rotation axis. We proposed a mechanism for the symmetry breaking by which rod-climbing-like effects due to nonlinear viscoelasticity around the rotating sphere provide a propulsive force by squeezing at the rear of the sphere. A theoretical analysis shows that such propulsive forces develop quite generally in a range of fluids

with nonlinear constitutive laws, and flow visualization reveals inward radial flows consistent with this mechanism. The two symmetry-broken states, and hence direction of propulsion, can be selected between by the application of a static magnetic field along the rotation axis, enabling guided 3D propulsion.

## Methods

**Rheology of nonlinearly viscoelastic fluids**. Human mucus can range in mucin glycoprotein concentration between 2% and 5%; we chose 4% as a baseline value[17]. Synthetic mucus was synthesized using mucin from porcine stomach (Sigma Aldrich, ME2378), by adding mucin to 150 mL of deionized water to produce a 4% mucin w/v synthetic mucus formulation. A stir bar and hot plate set to 60 °C was used to agitate the mixture for 30 min. The mucus was then transferred to three 50 ml centrifuge tubes and centrifuged for 10 min at 1200 relative centrifugal force (rcf) to remove any large aggregates (>5 μm) of undissolved mucus particulates from the sample. The supernatant was then transferred to fresh tubes and stored at 4 °C until used in experiments. Scanning electron microscopy (SEM) was used to

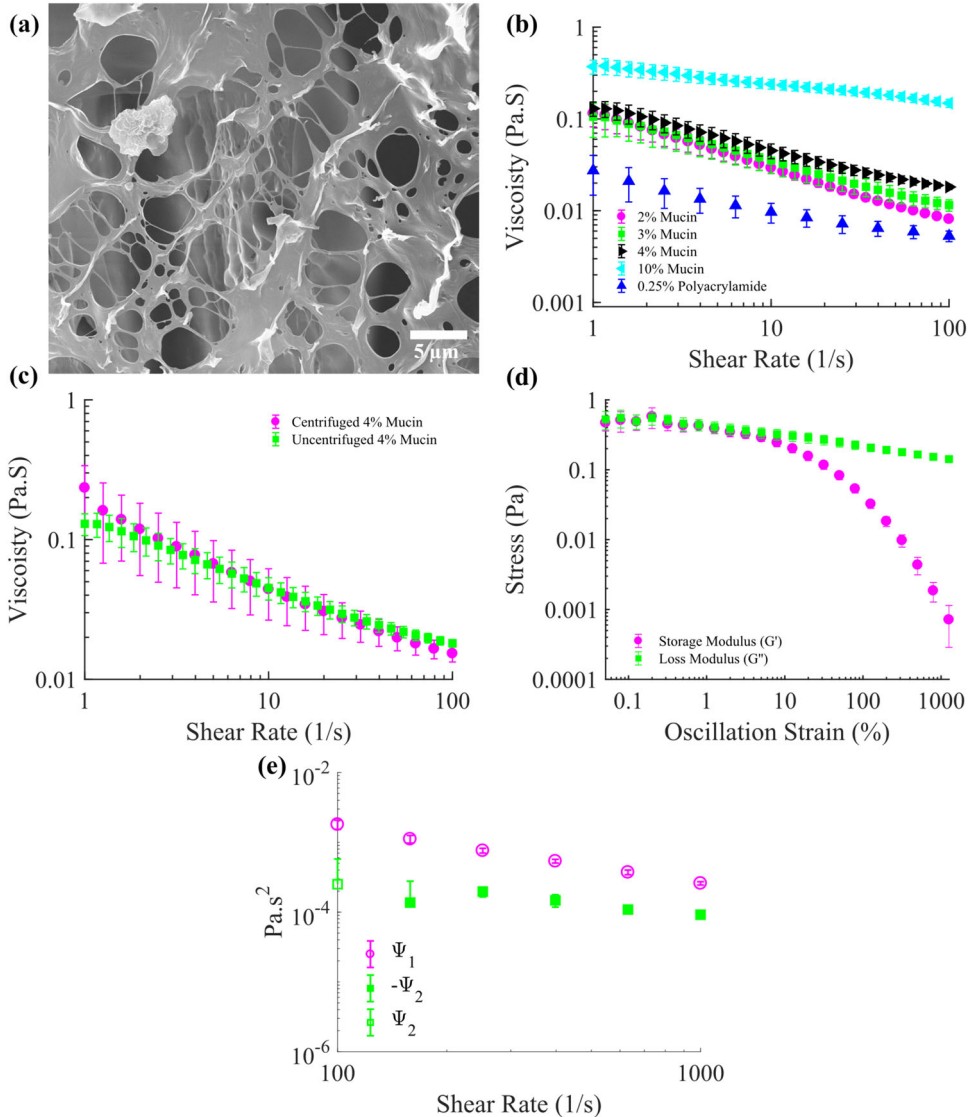

**Fig. 5 Rheological characterizations of synthetic mucus and polyacrylamide. a** Scanning electron microscopy (SEM) image of synthetic mucus; similar images were produced in two other mucin samples. **b** Viscosity vs. shear rate curves for different concentrations of mucin and 0.25% polyacrylamide. **c** Comparison of centrifuged and uncentrifuged 4% mucin formulations. **d** Evolution of storage modulus (Pa) and loss modulus (Pa) with increasing oscillation strain for 4% mucin; frequency was fixed at 10 Hz during this experiment. **e** First ($\psi_1$) and second ($\psi_2$) normal stress coefficients for 10% mucin with theoretical inertial correction (see Supplementary Note 10). Three independent trials were performed for individual concentrations in (**b**), three independent trials were used for data shown in (**c**, **e**), and four independent trials were performed for (**d**). Data are presented as mean ± standard deviation. Source data are provided as a source data file.

visualize the heterogeneous mesh fibers and the porous voids that make up mucins' overall structure (Fig. 5a). Using a TA Instruments Discovery Hybrid Rheometer (DHR-3) the mucus samples were characterized using stress sweeps, which were performed using a 40 mm parallel plate under an incremental shear rate from 1 to 100 (1/s) with 30 s of averaging time used at each data point, with 5 points collected per decade. The 0.25% polyacrylamide samples were characterized using stress sweeps, which were performed using a cone-and-plate (40 mm diameter, 4° angle) geometry under an incremental shear rate from 1 to 100 (1/s) with 30 s of averaging time used at each data point, with 5 points collected per decade. The averaged results for different mucin concentrations (at least three trials each) and 0.25% polyacrylamide are reported in Fig. 5b where all samples possessed a clear and consistent shear thinning effect. As the concentration of mucin increased, the overall viscosity increased monotonically and the resulting viscosity curves were in good agreement with previous experimental studies reported in literature[45]. The results for 0.25% polyacrylamide were also similar to those reported in literature[19]. Additionally, it was found that the centrifugation step only moderately affected the overall rheological behavior of the mucus, as can be seen in Fig. 5c which compares centrifuged and uncentrifuged 4% mucin formulations. Interestingly, mucin concentration was also demonstrated to positively effect microparticle propulsion as concentration increased (see Supplementary Note 9). Oscillation experiments performed on the DHR-3, using a 40 mm parallel plate with a frequency of 10 Hz,

for 4% mucin revealed that at low oscillation strains the storage and loss moduli (Pa) had similar magnitudes (Fig. 5d). As the oscillation strain increased the storage modulus decayed sharply, it resulted in the mucus behaving more like a liquid once the loss modulus was dominant; this is consistent with the documented viscoelastic behavior of biological mucus[17]. Biological human mucus was not used in experiments because it had inconsistent rheological properties (which vary between humans) and was difficult to store for long periods of time. While this synthetic mucus formulation lacks many components typically found in human mucus (lipids, salts, DNA, proteins, etc.), literature suggests that mucus viscoelastic behavior is completely dominated by mucin glycoprotein concentration, whereas the other components only marginally contribute to these properties[17]. The saline solution used in control experiments was created by mixing deionized water and NaCl (Sigma Aldrich, S5886) in a 15% w/v ratio. Polyacrylamide solutions were fabricated by mixing polyacrylamide (Sigma Aldrich, 92560) with deionized water in a 0.25% w/v ratio. The polyacrylamide mixture was agitated overnight using a magnetic stir bar and heated at a temperature of 30 °C. Methylcellulose (Sigma Aldrich, M0512) at a 0.2% concentration was synthesized using directions provided by Sigma Aldrich. It is known that polyacrylamide solutions develop first and second normal stress differences under applied shear[20,21]. We also measured the first and second normal stress differences developed by high-concentration mucin solutions under shear from the axial forces measured by a TA Instruments AR-G2

rheometer using cone-and-plate (60 mm diameter, 2° angle) and parallel plate (60 mm diameter) geometries[57,58] (see Supplementary Note 10 for details). The measured first and second normal stress coefficients for 10% mucin solution are shown in Fig. 5e, with a positive first normal stress coefficient and smaller negative second normal stress coefficient. Such normal stress differences are known to lead to rod-climbing effects. We also measured first and second normal stress differences in 2.5% polyacrylamide with results comparable to literature (Supplementary Note 10). Note that the strain rate expected nearby a rotating sphere is equal to the angular velocity of the sphere; for our experiments this varies from 0 to 120 s$^{-1}$.

**Magnetic field generating system and experimental setup.** Avidin-coated 10.6 μm ferromagnetic particles (Spherotech, SVFM-100-4) were suspended inside synthetic mucus, or polyacrylamide solutions, and were used for the majority of the experiments. Other microparticles utilized included 2 μm (SVFM-20-5), 4 μm (SVFM-40-5), and 8 μm (SVFM-80-4) diameters. Samples were prepared such that 0.5–1 μl of the ferromagnetic particle suspension, of the selected diameter, was added to 1.5 ml of either mucus or polyacrylamide solution. The sample was then vortexed slightly and placed on a permanent magnet for 15 s to magnetize the particles. Afterwards, 40 μl of the sample was added to a polydimethylsiloxane (PDMS) chamber, approximately 2 mm in diameter and 1 mm in height, and situated on a number 1.5 glass coverslip (25 × 30 mm$^2$). An 18 × 18 mm$^2$ coverslip was then placed on top of the chamber to seal it, minimizing both evaporation and internal flows. The chamber was then loaded into an approximate Helmholtz coil system mounted to a Leica DM IRB inverted microscope (Type 090-132.701); a full description of the approximate Helmholtz coil system and it's magnetic field profiles can be found in Supplementary Note 7. The Helmholtz coils do not produce any significant magnetic field gradients, and Supplementary Note 8 demonstrates that only rotating magnetic fields can propel a microparticle. A 63× objective was used to visualize microparticle motion within the sample chamber. A complementary metal oxide semiconductor (CMOS) camera (Point Grey, FL3-U3-13Y3M-C) was used to record experiments at 30 frames per second (fps) with a pixel resolution of 512 × 640 (0.152 μm/pixel). A customized LabVIEW program was developed to control signal outputs to interfaced digital acquisition (National Instruments, DAQ) control boards, which then directed those signals to the attached power supplies (KEPCO, BOP-20-5M). There was one power supply per coil pair within the approximate Helmholtz coil system. The magnetic fields generated by this system are mostly uniform and no meaningful magnetic field gradients are produced. A magnetic field was generated that is described by,

$$\mathbf{B} = \begin{bmatrix} -B_s \cos\theta + B_r \sin\theta \cos\omega t \\ B_s \sin\theta + B_r \cos\theta \cos\omega t \\ B_r \sin\omega t \end{bmatrix}, \tag{2}$$

$$\mathbf{n} = \begin{bmatrix} -\cos\theta & \sin\theta & 0 \end{bmatrix}, \tag{3}$$

where $B_s$ is the static magnetic field amplitude, $B_r$ is the rotational magnetic field amplitude, $\omega$ is the frequency (rad/s), $t$ is the time (seconds), and $\theta$ is the heading angle of the propulsion direction. In experiments, $B_r$ was adjusted to be proportional to the applied frequency ($|B_r| = 0.5f$, where $f$ is the frequency in Hz) in order to prevent the magnetic microparticle rotation from desynchronizing with the rotation of the magnetic fields[5]. The propulsion direction is defined to be along the rotation axis and corresponds to the heading vector (**n**). When looking from behind the heading vector (**n**), rotation can be defined as counterclockwise with positive $\omega$ and clockwise with negative $\omega$. In a Newtonian fluid, when there is no static field ($B_s = 0$) and the rotational frequency is below a critical step-out frequency, a spherical microparticle with a permanent dipole moment rotates about the heading vector (**n**) at the same rotational frequency as the field, and its dipole moment is in the plane perpendicular[6] to **n**. When there is an additional static field, the moment is tilted at a constant angle out of the plane perpendicular to **n**, and the moment and microparticle rotate about **n** at the same frequency as the field, which makes the moment co-rotate with the field. This rotation is combined with an additional rotation of the particle about the moment direction (see Supplementary Note 5).

**Experimental procedures.** The first set of experiments involved rotating microparticles suspended within fluids under incrementally increasing rotational frequencies. The rotation axis (**n**) and propulsion direction were set to be along the positive x-direction, with the z-direction (height) oriented to be perpendicular to the coverslip. The transverse velocity was found to be a function of distance (Supplementary Figs. 1b and 2) from the substrate of the sample chamber and rotation direction (Supplementary Fig. 3) of the rotating magnetic field; this motion corresponds to well-understood rolling motion along the nearest boundary[15,16]. As mentioned previously, in this paper we focus on propulsion along the rotation axis (**n**), which is related to the fore–aft symmetry. For the experiments in Fig. 1, the static field ($B_s$, Eq. (2)) was set to 2 mT, the heading angle ($\theta$) was set to 0°, while the frequency ($\omega$), was incremented from +1 to +18 Hz at 1 Hz increments ($\omega = 2\pi f$, where $f$ is the frequency in Hz) counterclockwise when viewed from behind the heading vector. The time between each increment during the experiments was variable, but the video frames captured by the CMOS camera during the experiments could be precisely correlated to each increment, allowing for the accurate extraction of velocity data. All experiments described involve particles >100 μm from boundaries and at least three

trials were conducted per microparticle with at least four microparticles examined per fluid; the results of all microparticles were then averaged together. Velocities were calculated by dividing the displacement of a particle between frames by the sampling time (1/30 s). While the velocities plotted in Fig. 1 represent the average of the data, it is important to note that microparticles moved faster or slower depending on uncontrollable factors (location in mucin, diameter variations, etc.) and leads to some differences in velocities for Fig. 3.

To test open-loop directional control of microparticles, the rotational frequency was fixed as a constant (counterclockwise when viewed behind the heading vector, see Fig. 2), $B_s$ was set to 2 mT, and $\theta$ was incremented by 90° at user-specified time intervals. The proportional controller for 2D closed-loop control used the same parameters for rotational frequency and $B_s$, but $\theta$ was governed by

$$\dot{\theta} = k\alpha_d, \tag{4}$$

$$\alpha_d = \varphi - \theta, \tag{5}$$

where $\dot{\theta}$ is the time derivative of the heading angle, $k$ is the gain parameter, $\alpha_d$ is the difference between the direction of the desired position relative to the microparticle ($\varphi$) and the heading angle of the microparticle ($\theta$). For all experiments, in order to ensure a fast response from the controller, $k$ was set to 5. More advanced proportional-integral-derivative (PID) controllers and 3D closed-loop controllers will be explored in future work.

For experiments involving static magnetic field incrementation, rotational frequency was fixed at 15 Hz (or 94.25 rad/s, unless specified otherwise), the rotation direction of the magnetic field was selected to be counterclockwise when viewed from behind the rotation vector, and $\theta$ was set to 0° to ensure propulsion along the x-direction. The static magnetic field, $B_s$, was swept between −5 and 5 mT in 0.2 and 1 mT increments; again, the times between these increments were variable but the individual frames of the recorded videos could be matched to the experimental parameters being incremented. Hysteresis experiments were performed at the same incrementation, but between 2 and −2 mT and then back to 2 mT. All microparticles were examined over a minimum of at least three trials.

The μPIV data were collected using a Nikon Eclipse TI inverted microscope with a 100× total internal reflection (TIRF) objective. An electron-multiplying charge-coupled device (EMCCD) camera (iXon 897, Andor Technology) was used to collect video captures at 60 fps. The experimental parameters were the same as the directional control experiments such that $B_s = 2$ mT and the frequency was 15 Hz under counterclockwise rotation (when viewed from behind the rotation vector). Tracer particles with 200 nm diameter (Thermo Scientific, Fluoro-Max, G200) were mixed in a ~0.2 mg/ml concentration with the mucus medium and dispersed through vortexing. Due to experimental limitations, most of these experiments were performed close to the boundaries of the sample chamber, which led to both exaggerated transverse rolling translation (Supplementary Fig. 3) and reduced propulsion velocity (Supplementary Figs. 1b and 2). The videos and extracted velocity flow fields were analyzed using LaVision DaVis PIV software. Individual videos were converted into a series of still images where each image represents a single frame in the video. First, a PIV time series operation was performed to create velocity vectors for each individual frame. The velocity vectors from the individual frames were averaged over the frames over the time periods of the videos up to a selected time (usually 20 s). Flows for the selected times of the experiments are presented in Fig. 4 and Supplementary Note 12; numerical information for all μPIV experiments is tabulated in Supplementary Note 13. Magnetic fields used to rotate the microparticles for these experiments were generated by a MagnebotiX magnetic field generator (MFG-100-i), which could produce rotating magnetic fields and static magnetic fields comparable to the approximate Helmholtz coil system discussed previously.

## Data availability
The source data underlying Figs. 1b, c, 3, and 5b, d, e, as well as Supplementary Figs. 1, 2, 3b, c, e, f, 4–7, 14–17 are all provided in Supplementary Data 1 with the supporting information; additional data sets for specific experiments (if available) will be provided upon request from the authors. Any other data sets presented in this manuscript and in the SI, but not included with the source data files because of their format complexity, are available upon request from the authors.

## Code availability
All code used for analysis in this manuscript is available on request from the authors.

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

## Acknowledgements

We would like to thank the National Science Foundation (CMMI: 1712096, 1760642, and 1761060; HRD: 1735968; DMR: 1644779 and 1455247) and the National Institutes of Health (1R01GM131408-01) for providing the resources to conduct these experiments. We thank Dr. U Kei Cheang and Dr. Hoyeon Kim for providing the ground work to conduct these experiments. We thank Micah Oxner, Benjamin Woodruff, Amanda Liew, and Richard Burns for conducting preliminary experiments. We thank Dr. Nuwan Bandara and Dr. Buddini Karawdeniya for their guidance in this manuscript's preparation. We thank Anuruddha Bhattacharjee for helping measure the magnetic fields of the Helmholtz coil system.

## Author contributions

L.W.R. conducted the experimental research involving microparticles, fluid fabrication/characterization, and motion analysis. J.A. conducted investigations into synthetic mucus mesh structures using scanning electron microscopy and provided guidance for mucus fabrication. X.Z. provided methods for 3D tracking of microparticles and measured the magnetic fields of the Helmholtz coil system. J.N.W. measured the normal stress differences in the fluids. H.C.F. was the co-principal investigator and developed the theoretical framework for spontaneous symmetry-breaking propulsion of microparticles. M.J.K. was the co-principal investigator for the experimental campaign and provided the resources necessary for all experiments.

## Competing interests

The authors declare no competing interests.
