## [Peer Review File · Nature Communications]

Reviewers' comments:

Reviewer #1 (Remarks to the Author):

This work describes both a theoretical and experimental investigation of symmetry-breaking propulsion of rotating spherical microparticles in a synthetic mucus medium.

Their suggested propulsion mechanism is based upon hoop stresses and nonlinear fluids with the generation of rotating flows around these spherical microparticles. It is this combination of a symmetric particle with the flow characteristics of the non-newtonian fluid which lead to motion.

The paper is clearly written, the data and experimental verification are convincing.

The work is to a high quality. The paper is well written and sufficient repeats all combine to produce a paper of the highest quality.

I have no hesitation in suggesting this work be published.

My only comment/concern is the method of magnetisation of the ferromagnetic core. The idea being that exposing the particle to a magnetic field for 15s imposed a degree of magnetism to the particle. I'd expect there to be a degree of variation in the final degree of magnetism to each particle which would contribute to the behaviour of the particle. Is there a way in which this could be quantified in some way ?

Additionally given the helmholtz coils are meant to be mounted a given distance apart, it would be informative to know the full set-up of the coil arrangement.

Reviewer #2 (Remarks to the Author):

The authors report that rotating spherical microparticles can propel in a synthetic mucus medium along the rotational axis. A static magnetic field can be applied to control the propulsion direction of the sphere. The authors try to explain the propulsion mechanism with reference to "spontaneous symmetry-breaking" based on "hoop stresses" in a nonlinear fluid. In general, the observed phenomenon is surprising and interesting, as it could be a very simple propulsion scheme for a microswimmer. However, many questions have not been addressed, there are major concerns regarding the explanation that has been provided, and more experimental and theoretical evidence should be provided to prove this new concept. My detailed comments follow:

1) The theoretical explanation is very vague in the paper and no results showing the quantitative agreement between the proposed theory and experimental results are given. The authors mentioned in the main text Page 10 Paragraph 2: "...fluid elements at the back of the sphere have spent more time circling the sphere than those at the front; thus, streamlines at the back have larger excess tension than streamlines at the front, producing a net squeezing force that propels the sphere forward." This is only a qualitative explanation without any simulation or experimental evidence. More evidence is needed to convince this reviewer or the general reader.

2) The "hoop stresses" and "non-linear fluid" are mentioned many times in the paper (also in the abstract), what is the definition of the hoop stress here? Which relationship is non-linear in the mucus solution, and which parameter (shear-thinning viscosity, elastic modulus or something else) is actually important for propulsion? Please clarify.

3) Related to the above point: Although some PIV results are included in Fig. 4, the authors fail to link their experiment to their theory, or to clearly explain the propulsion mechanism. One major improvement of the paper could be if they use numerical simulations to simulate the flow field and the normal stress, and the former should be quantitatively compared with the PIV results. The PIV result is close to the hard boundary (Page 20 Line 1), how do you prove that it really reflects the flow in the bulk?

4) The experimental results seem very irreproducible. For example, in Fig. 8(c) the four particles show very

different behaviors, and the propulsion velocity in Fig. 8(a) and (b) are inconsistent, i.e. four times difference for the top velocity. The authors attribute this to many reasons including static field history and increment size, individual particle variation, and the mucus environment (Page 9 Paragraph 1). I do not think these are valid explanations. The phenomenon is not well-enough understood and the authors should provide better data.

5) The authors tried to explain the phenomenon related to the "rod-climbing" effect (Page 10 Paragraph 2), which is due to the normal stress differences induced by shear stress. It is generally a good idea to use this effect to propel a microswimmer. However, to fully understand it, the rheological results of the normal stress difference are missing. More rheological measurements of the fluid showing the first and second normal stress differences should be included in the paper, following the established method in the literature, e.g. Mall-Gleissle et al. The normal stress behaviour of suspensions with viscoelastic matrix fluids, *Rheol Acta* (2002) 41: 61–76.

6) Actually, what is the reason to choose the mucus solution in the study, why not use some model fluids that are well characterized and that show significant "rod-climbing" effects to clearly illustrate the propulsion mechanism?

7) The propulsion velocities of different sized particles cannot be rationalized and need more experiments and explanations. The velocity clearly increases with sizes above 4-micron-diameter (Fig. 1(c)), it is contradictory with the size-dependent propulsion in porous media (what the authors wrote "smaller microparticles' ability to move through the voids commonly found in mucus structures" in Page 5 Last Line). Particles at larger size than the pore size should not move (or move at lower speed), and the pore size seems to be around 4-5 microns according to SEM (Fig. 5(a)). Should not the "transverse velocity" increase with the rotational frequency (Fig. S1(a) purple symbols), if the particle is touching the bottom as a rolling "wheel" (Page 4, third to the last line)?

8) The magnetic gradient of the coil system used in the study must be carefully measured in all directions and clearly reported. Even Helmholtz coil set-ups have inhomogeneities in the center of the field. Control experiments should be performed to prove that the propulsion has nothing to do with any magnetic gradient.

9) Why was a 4% mucus solution selected for the experiment? What are the propulsion results for other concentrations?

10) Why do ferromagnetic particles need to be placed on a permanent magnet for magnetization (Page 16 Line 6)? Does it result in the agglomeration of the ferromagnetic particles?

Reviewer #3 (Remarks to the Author):

This manuscript describes a counter-intuitive form of propulsion, whereby the fore-aft symmetry of a rotating sphere is broken dynamically by the elasticity of the fluid medium. Viscoelastic media are important in biomedical contexts, especially, but propulsion in these media is studied much less than in Newtonian fluids, due in part to the difficulty of modeling their mechanics. The form of propulsion in the present experiments is, to my knowledge, unlike all others in microorganisms or considered for micro-robotics, wherein the propeller geometry and/or the driving select a direction; it appears to be a first for low-Reynolds-number propulsion. In these experiments the propulsive force is due to the nonlinear elastic hoop stresses generated around the rotating sphere, a mechanism that has been demonstrated elsewhere. What is new is that this force has a net direction not because of geometry, but because of the sphere's ongoing translation. The authors use the ubiquitous physics paradigm of spontaneous symmetry breaking, and show, fittingly, that a static magnetic field breaks the symmetry deterministically and selects the direction of motion. The supplemental information is thorough and useful. While it's unclear whether this form of propulsion is efficient and responsive enough to find practical applications in micro-robotics, drug delivery, etc., these experiments isolate an effect that is likely at play in more common and complicated forms of artificial and biological propulsion in viscoelastic fluids. This work has the potential to be of great interest to the broader propulsion and complex fluids communities.

My foremost concern has to do with justifying the choice of fluids in these experiments. The use of artificial mucus as the medium seems to needlessly complicate the experiments and detract from the proposed

explanation of this newly-observed phenomenon, because these fluids have some structure on the scale of the spheres and their trajectories, and have unusual interactions related to their biological function. Mucus seems more appropriate for a proof-of-concept for applications, rather than a systematic study to elucidate the mechanism. Why did the authors choose mucus, and would these experiments have also worked with polymer that is not so specialized, such as polyacrylamide?

Similarly, the authors seem to be very reluctant to identify nonlinear fluid elasticity as the key ingredient that makes this form of propulsion possible—it is not mentioned in the abstract or introduction, even at the mention of [elastic] hoop stresses. This is confusing and makes one wonder whether shear-thinning or any of the special additional properties of mucus (related to its biological function) are important or sufficient for propulsion. I think that this choice of emphasis does a disservice to the modeling component of the paper, and again detracts from this work's impact.

Finally, it was unclear why the authors did some experiments with a methycellulose solution, or how the results support their conclusions. Why was it important to have a second Newtonian system—especially a complex fluid that is only approximately Newtonian?

I have a handful of more substantive concerns and suggestions that the authors should consider in their revisions, as well as several more minor issues.

- The authors are unfortunate in that ferromagnetism has some of the same behaviors as they observe for propulsion, and this will probably occur to many readers. Specifically, the text needs to identify why these particles' magnetic hysteresis cannot be the origin of the hysteresis in Fig. 3d. (Variability in magnetic permanence is, at least superficially, an alternate explanation for Fig. 3c.)
- The supplemental information details a model based on a general framework by Giesekus. This earlier work shows the possibility of a propulsive force and spontaneous symmetry-breaking, albeit for an idealized fluid and for perturbative velocities. I think that this is worth mentioning in the introduction of the main text. It will hardly detract from the novelty of the present work, and it will increase its significance by making readers more confident that this possibility arises from generic nonlinearities of viscoelastic fluids but was hiding in plain sight.
- The model appears to show plainly that a rotating and non-translating sphere ($\Omega > 0, U = 0$) can be linearly unstable to translation. If true, this is worth pointing out in the main text, since it directly leads to spontaneous symmetry-breaking, and is the crux of a dynamical analysis of the system (see e.g. Strogatz, "Nonlinear Dynamics and Chaos").
- Can the authors rule out magnetic interactions between spheres in their experiments? Each micrograph shows only one sphere, but it's unclear how that was achieved.
- It seems that the experimental protocols implicitly rules out the important possibility that magnetic field gradients are causing the appearance of propulsion. The authors should consider making this explicit. Additionally, it's unclear why the motion of a sphere without actuation that the authors observe is attributed to convective flow and not field gradients.
- I may have missed it, but it seems like there is very little effort to connect the measured rheology of these fluids with strain rates in the experiments. Could the authors comment on the shear strain rate regime in which these experiments take place?

Minor issues

- In Figs. 1, 2, 4, and S2, the authors should strongly consider replacing the "jet" colormap with a perceptually uniform one, which is available in all major scientific plotting packages. Perceptually uniform colormaps have been the best practice for roughly the past decade, since they are easier to interpret quantitatively, and as a bonus, do not needlessly hinder readers who are colorblind or who have black-and-white printers.
- Paragraph beginning at bottom of p. 7: In this paragraph I was confused about the direction of the static magnetic field and how it related to the propulsion direction. It's also unclear why "(mT)" is included in the first sentence of the paragraph.
- Caption of Fig. 3(a): Range should be -5 to 5 mT.
- p. 9: At the beginning of this paragraph, in "To understand how a static field might be able to control the propulsion direction", "understand" seems like a poor word choice since this remains a major open question.
- Top of p. 11: Should be "Stokes' Law" or "Stokes's Law". Or one could write "Stokes drag on a translating

sphere.”

- In Fig. 4, why not rotate (a) or (b) so that they have the same propulsion direction? The figure would then better support the comparison in the text.
- It’s appropriate in Fig. 4 to scale velocities by the maximum velocity in each experiment, as the authors do. However, this leaves the reader without a sense of how these velocities compare with the spheres’ translation velocities that are the primary concern of the paper. Can this information be included?
- Can the authors comment further on the apparent super-linear increase in propulsion velocity with sphere size? What would limit the propulsion of a mm-scale sphere, or larger?
- On p. 16, at “the ferromagnetic particle solution,” “suspension” would be a much better term.
- In the “Contributions” section (p. 20), “principle investigator” should be “principal investigator” (2 instances).

Reviewers' comments:

Reviewer #1 (Remarks to the Author):

This work describes both a theoretical and experimental investigation of symmetry-breaking propulsion of rotating spherical microparticles in a synthetic mucus medium.

Their suggested propulsion mechanism is based upon hoop stresses and nonlinear fluids with the generation of rotating flows around these spherical microparticles. It is this combination of a symmetric particle with the flow characteristics of the non-newtonian fluid which lead to motion.

The paper is clearly written, the data and experimental verification are convincing.

The work is of a high quality. The paper is well written and sufficient repeats all combine to produce a paper of the highest quality.

I have no hesitation in suggesting this work be published.

My only comment/concern is the method of magnetisation of the ferromagnetic core. The idea being that exposing the particle to a magnetic field for 15s imposed a degree of magnetism to the particle. I'd expect there to be a degree of variation in the final degree of magnetism to each particle which would contribute to the behaviour of the particle. Is there a way in which this could be quantified in some way ?

We thank the referee for the positive feedback and for the suggestions. It is common practice to magnetize ferromagnets by applying a strong magnetic field and in our prior work, and the work of other groups, this is also commonly done for magnetic microrobots. Unfortunately, quantifying the magnetism of each particle would be a challenging endeavor, as the placement of the microparticles on the magnet varied between different experiments. Fortunately, as long as each microparticle has a permanent magnetic moment, no matter its magnitude, the same behavior is expected for small rotation frequencies (below the step-out frequency, see Methods p. 21). The direction of magnetization does not affect the results due to the spherical symmetry of the bead.

Additionally given the helmholtz coils are meant to be mounted a given distance apart, it would be informative to know the full set-up of the coil arrangement.

We originally excluded this information and referred to previous work. However, we have now added a detailed description of the system in the SI during revision, which also includes characterization of the magnetic fields produced by the coil system for all 3 coil pairs (SI section 7).

Reviewer #2 (Remarks to the Author):

The authors report that rotating spherical microparticles can propel in a synthetic mucus medium along the rotational axis. A static magnetic field can be applied to control the propulsion direction of the sphere. The authors try to explain the propulsion mechanism with reference to “spontaneous symmetry-breaking” based on “hoop stresses” in a nonlinear fluid. In general, the observed phenomenon is surprising and interesting, as it could be a very simple propulsion scheme for a microswimmer. However, many questions have not been addressed, there are major concerns regarding the explanation that has been provided, and more experimental and theoretical evidence should be provided to prove this new concept. My detailed comments follow:

1) The theoretical explanation is very vague in the paper and no results showing the quantitative agreement between the proposed theory and experimental results are given. The authors mentioned in the main text Page 10 Paragraph 2: “...fluid elements at the back of the sphere have spent more time circling the sphere than those at the front; thus, streamlines at the back have larger excess tension than streamlines at the front, producing a net squeezing force that propels the sphere forward.” This is only a qualitative explanation without any simulation or experimental evidence. More evidence is needed to convince this reviewer or the general reader.

We have made the following major changes to the manuscript to address your comments:

1. We have clarified the description of the theoretical explanation to make it more precise (see reply to comment 2 for details).

2. In the original manuscript, the symmetry breaking mechanism was supported by the theoretical perturbative calculation. In addition, we noted that our PIV results showed significant radial flows around the propelled spheres, which corroborates the theory since the secondary flows predicted by theory adds radial components to the flow, which would be purely azimuthal in a Newtonian fluid. We have now added a comparison between the PIV flow and the predicted secondary flow (see reply to comment 3 for details).

3. As suggested by the referee, we have performed additional experiments in another fluid, a low concentration polyacrylamide solution, which is known to show rod-climbing effects. We have observed symmetry breaking propulsion in this fluid as well, further bolstering our conclusions (see reply to comment 5 for details).

2) The “hoop stresses” and “non-linear fluid” are mentioned many times in the paper (also in the abstract), what is the definition of the hoop stress here? Which relationship is non-linear in the mucus solution, and which parameter (shear-thinning viscosity, elastic modulus or something else) is actually important for propulsion? Please clarify.

We agree with the referee that it would be better to be more precise with our explanation of the physics behind the observed phenomenon. In the original manuscript we used the term “hoop stresses” to refer to the heuristic physical explanation of effects similar to the rod-climbing effect, in which excess tension (the circumferential hoop stress) along circular streamlines surrounding rotating objects squeezes the inner fluid upwards. In terms of nonlinear viscometric

quantities, the first normal stress difference corresponds to this hoop stress. However, for our propulsion mechanism, axial forces caused by a rotating flow in a nonlinear fluid can arise due to both first and second normal stress differences; indeed, the surface deformation for rod-climbing in a second order fluid depends on both the first and second normal stress coefficients (see Ref [21], Bird, Armstrong, Hassager, p 330). Thus it is not enough to refer to only hoop stresses in our explanation of the phenomenon, as we did in the original manuscript, and ignore the other components of stress. Generally, however, these physical effects arise from the elastic response of polymer molecules or mucin in solution, so instead we now refer to them as nonlinear viscoelastic effects.

Furthermore, our theoretical analysis shows that more than the first and second normal stress coefficients are needed to correctly describe propulsion: the first and second normal stress coefficients serve to define a second order fluid, but since the propulsive force is a third order effect in the retarded motion expansion, a consistent description also depends on the third order fluid parameters in addition to the first and second order normal stress coefficients. The form of the propulsion force shows that the propulsion mechanism depends on all the nonlinear parameters of the third order fluid, which generally impact (in complicated ways) observed phenomena such as those mentioned by the referee (shear-thinning viscosity, elastic modulus). Thus the mechanism cannot be attributed to a single one of these nonlinear effects.

We thank the referee for pushing us to be more clear with our explanation. To more precisely reflect the mechanism, we have replaced the description of the mechanism in the abstract with

We propose and experimentally corroborate a propulsion mechanism for these spherical microparticles, the simplest microswimmers to date, arising from nonlinear viscoelastic effects in rotating flows similar to the rod-climbing effect.

In the introduction on p. 3 we have added the following description:

We propose a physical mechanism for symmetry breaking that arises from nonlinear viscoelastic effects in rotating flows, similar to the rod-climbing effect, which pushes the sphere along its rotation axis.

In addition, on p 11, we have rewritten the explanation of the propulsion mechanism as follows:

we propose that the symmetry breaking mechanism is caused by nonlinear viscoelastic stresses that arise in rotating flows, similar to those responsible for the “rod-climbing” effect. Around a vertical rotating rod in a polymeric fluid, nonlinear first normal stress differences cause excess circumferential hoop stresses along the circular streamlines, while second normal stress differences cause excess radial stress; both can squeeze the fluid upwards around the rod²¹. We note that both polyacrylamide and mucin solutions display first and second normal stress differences that lead to rod-climbing-like effects (see Methods, Fig. 5 and SI section 10). For a rotating and translating sphere, fluid material also advects past the sphere, so that fluid elements at the back of the sphere have spent more time circling the sphere than those at the front; thus, the back of the sphere has larger nonlinear viscoelastic stresses than the front, producing a net squeezing force that propels the sphere forward. A force-free symmetry-breaking translational state occurs when this propulsive force balances the drag force from translation.

Then, at the end of the next paragraph (p. 12), after describing the perturbative results for the propulsive force, we have added

Finally, we note that although in a second-order fluid, rod climbing is directly related to the first and second normal stress coefficients that specify the constitutive law²¹, our propulsive force is a higher order effect that can only be consistently described using all the parameters that specify a third order fluid, implying that nonlinearities beyond the normal stress differences impact symmetry breaking propulsion.

We note that polyacrylamide solutions are known to develop first and second normal stress differences under shear, and that we have now measured the first and second normal stress differences in 10% mucin solution. These nonlinear viscoelastic effects lead to rod-climbing-like effects which should be present in our fluids (also see response to comment 5).

3) Related to the above point: Although some PIV results are included in Fig. 4, the authors fail to link their experiment to their theory, or to clearly explain the propulsion mechanism. One major improvement of the paper could be if they use numerical simulations to simulate the flow field and the normal stress, and the former should be quantitatively compared with the PIV results.

As the referee suggested, we have now added a comparison between experimental PIV results and theory to directly tie them together. While in a Newtonian fluid, the flow around a rotating sphere is purely azimuthal, the PIV results for a propelling sphere also show radial flows. The radial component is a direct result of the nonlinear viscoelastic effects and is expected from our propulsion mechanism. As we now state on p 14 of the revised manuscript:

When examining a microparticle propelling in the positive z -direction, we found that in addition to a persistent azimuthal component [Fig. 4 (h)], there is a radial component [Fig. 4 (g)] that converges towards the microparticle, resulting in the velocity flow field shown in Fig. 4 (i). Notably, in contrast to the flow around a rotating sphere in a Newtonian fluid, which is purely azimuthal²⁴, the significant inward radial flow is consistent with the secondary flows expected from our hypothesized propulsion mechanism involving rod-climbing-like effects. Supporting our proposed flow mechanism, we found that the observed radial flow closely agreed with the secondary flow predicted by the theoretical perturbation expansion described above (see SI Section 11 for details).

We note that this since the theory is perturbative, in an important sense this comparison is still fundamentally qualitative since it does not access the strongly nonlinear regime. As the referee suggests, a full numerical simulation would be required to perform a fully quantitative comparison with the experimental results. However, we have not performed this numerical solution for the following reasons:

1. A full numerical simulation would be quite difficult: while simulations using methods that could be applied to our particles in nonlinear fluids have been reported in literature (e.g., Li *et al.*, DOI: 10.1098/rsif.2017.0289; Binagia *et al.* DOI: <https://doi.org/10.1017/jfm.2020.456> and <https://doi.org/10.1039/C8SM02518E>), they are at the current state-of-the-art, and would be an entire project in themselves and therefore beyond the scope of this paper.

2. More importantly, the constitutive law of the fluid is not accurately known (particularly for mucus). For example, even in a perturbative treatment, the multiple parameters of a third order fluid are difficult to accurately determine experimentally since they would rely on fitting to behavior more complicated than the rheometric characterization of normal stress differences. This uncertainty could not be neglected since the propulsive force depends on all of them (see response to point 2). The existing numerical solutions referenced above typically choose a constitutive law (such as Oldroyd-B fluid). Doing the same in our case would lead to yet another qualitative, not quantitative comparison, hence we do not anticipate that they would add anything significant to the paper.

Due to the uncertainty in constitutive modeling, we emphasize to the referee that we have tested a wide range of the constitutive laws in the theoretical analysis, and they all show the symmetry-breaking propulsive force, which supports the generality of this physical phenomenon. In the revised manuscript, this theoretical evidence is further bolstered by the observation of symmetry breaking propulsion in a different fluid (polyacrylamide solution) with presumably different constitutive law (see response to comment 5).

The PIV result is close to the hard boundary (Page 20 Line 1), how do you prove that it really reflects the flow in the bulk?

We note the following which lends credence to the idea that the propulsion effect is not strongly affected by the boundary:

a) The 3D experiments shown in the SI video were all performed in bulk fluid media, with the exception of the second video in mucus, which was momentarily close to the boundary, and all three videos showing 3D experiments exhibited similar propulsive behavior.

b) In addition, in SI Section 1 we showed that proximity to the boundary only increased transverse velocity while still allowing some propulsive velocity.

c) For the particle in the PIV experiments [Fig. 4 (c) in original manuscript, now Fig. 4(g-i)] we have added to the end of the SI-Video included with this manuscript, showing that the microparticle actually went further into the bulk fluid, by ~ 20 microns, as rotation persisted (see SI Section 4). During this time, the radial secondary flow was always present and became more pronounced as the particle moved farther from the boundary. Thus, this characteristic of the flow, which we focused on, is not simply due to boundary effects.

d) We also performed μ PIV for microparticles in polyacrylamide that showed comparable results to the ones originally presented in 4% mucin. Additionally, a $2\ \mu\text{m}$ diameter microparticle in 0.25% polyacrylamide that was not near the substrate ($\sim 20\ \mu\text{m}$ away) also demonstrated similar inward radial flow behavior as it propelled (see SI Section 13).

4) The experimental results seem very irreproducible. For example, in Fig. 8(c) the four particles show very different behaviors, and the propulsion velocity in Fig. 8(a) and (b) are inconsistent, i.e. four times difference for the top velocity. The authors attribute this to many reasons including static field history and increment size, individual particle variation, and the mucus

environment (Page 9 Paragraph 1). I do not think these are valid explanations. The phenomenon is not well-enough understood and the authors should provide better data.

We agree with the referee that there is significant interparticle variation in the velocities which is not understood. However, the main focus of this paper is the existence of symmetry-breaking propulsion of spherical beads, which at a qualitative level is both surprising and highly reproducible. In mucin nearly all beads (90%) displayed propulsion and all of the propulsive beads tested for hysteresis displayed the pair of symmetry broken states at zero static field. In polyacrylamide, nearly all 2 micron beads (90%) and greater than 60% of 10 micron beads displayed propulsion with a static magnetic field, and every propulsive bead that we tested for hysteresis displayed the pair of symmetry broken states at zero static field.

We also point out that the quantitative behavior of each individual particle is reproducible between experiments, as shown in the SI Fig. S4 (a), in which the trials of individual microparticles displayed less variation than experiments involving different particles. Additionally, in the new experiments involving 0.25% polyacrylamide there is little interparticle variation [see Fig. 3 (d) and SI Section 2].

In the revised manuscript, we now emphasize the robustness of the propulsion behavior and provide more context about what fraction of particles were observed to propel, and of those, which displayed pairs of symmetry-broken propulsive states through hysteretic behavior:

p 5: Propulsion was ubiquitous among rotated microparticles; it was observed for beads of several different diameters (2, 4, 8 μm) in mucin solutions of various concentrations (2%, 3%, 4%) and the polyacrylamide solution (see SI Section 9). Propulsion was observed for >90% of beads in mucin solution, and >60% of beads in polyacrylamide solution.

P9: We were consistently able to demonstrate symmetry broken propulsive states in microparticles that showed propulsion. In mucin solution, all particles tested showed hysteresis when subjected to negative and positive sweeps of the magnetic field. In polyacrylamide solution, all 4 particles tested showed hysteresis (see SI Section 2, Fig. S6), although we note that the strength of hysteretic effects was dependent on frequency; when actuated at 15 Hz very little hysteresis was observed and microparticles switched propulsion direction at close to zero static field for both positive and negative sweeps.

The referee is correct that the variability in both propulsion velocity, and the strength of hysteresis is still not well-understood, but we regard quantitative understanding of the variability as out of the scope of our manuscript, which focuses on the surprising qualitative existence of propulsion. Given the inhomogeneity of the fluids, especially mucin, we also do not think it is surprising that variability is observed. In the original manuscript, we suggested a number of possible explanations for this variability in behavior. Upon reflection, we do not mean to say that these are the definitive explanation for the variability, so we have amended the statement to highlight that they are merely possible reasons for the behavior, which requires further study to fully understand on p. 8 of the main text:

Thus, we consistently observe the existence of propulsion and switching of propulsion direction as the static field is swept, although there is strong variability in the propulsion velocity and switching frequency for different static field histories and increment sizes (see

SI Section 2), which could be due to factors such as interparticle variation and medium heterogeneity (see SI Section 9). In the remainder of the paper we focus on understanding the robust existence of propulsion and whether the static magnetic field is necessary for propulsion to occur.

In addition, in the discussion section on p. 15 we now highlight how quantitative understanding of these issues still remains to be studied:

In this manuscript we focused on the existence and mechanism for symmetry breaking propulsion; further work is needed to completely understand the conditions (geometry, rotation frequency, fluid properties) required for symmetry breaking to occur, the size dependence and effect of boundaries of propulsion velocities, as well as how fluid properties and actuation history control the hysteresis and selection of propulsion direction.

We note that the variability of the switching behavior despite the robustness of propulsion was in fact a key clue that led us to suspect an underlying symmetry breaking as the propulsion mechanism -- i.e., the variability suggests that all these other factors are not crucial to establish the existence of propulsion. We now emphasize this point on p 9:

due to the robust observation of propulsion combined with variability in switching of propulsion direction we hypothesized that there is an underlying spontaneous symmetry-breaking translation

5) The authors tried to explain the phenomenon related to the “rod-climbing” effect (Page 10 Paragraph 2), which is due to the normal stress differences induced by shear stress. It is generally a good idea to use this effect to propel a microswimmer. However, to fully understand it, the rheological results of the normal stress difference are missing. More rheological measurements of the fluid showing the first and second normal stress differences should be included in the paper, following the established method in the literature, e.g. Mall-Gleissle et al. The normal stress behaviour of suspensions with viscoelastic matrix fluids, *Rheol Acta* (2002) 41: 61–76.

We thank the referee for this suggestion and reference. We have now included additional results characterizing the normal stress differences of 10% mucin solution in Fig. 5 of the manuscript and provided additional information in Section 10 of the SI. Polyacrylamide solution is well known in literature to possess first and second normal stress differences and we have added the following on p. 5 to the manuscript:

Polyacrylamide solutions are known to have first and second normal stress differences^{20,21}, and we were able to measure normal stress differences in 10% mucin solutions (see Methods, Fig.5 and SI Section 10).

We have also added this description to p.19 of the manuscript summarizing our results:

It is known that polyacrylamide solutions develop first and second normal stress differences under applied shear^{20,21}. We also measured the first and second normal stress differences developed by high concentration mucin solutions under shear from the axial forces measured by an TA Instruments AR-G2 rheometer using cone-and-plate (60 mm diameter, 2° angle) and parallel plate (60 mm diameter) geometries^{57,58} (see SI Section 10 for details). The measured first and second normal stress coefficients for 10% mucin solution are shown in Fig 5 (e), with a positive first normal

stress coefficient and smaller negative second normal stress coefficient. Such normal stress differences are known to lead to rod-climbing effects. We also measured first and second normal stress differences in 2.5% polyacrylamide with results comparable to literature (SI Section 10). Note that the strain rate expected nearby a rotating sphere is equal to the angular velocity of the sphere; for our experiments this varies from 0-120 s⁻¹.

The force transducer on both rheometers used in this manuscript did not possess the sensitivity necessary to reliably characterize the normal stress differences of 4% mucin and 0.25% polyacrylamide. Based on our results though we can infer that these first and second normal stress differences are most likely present in the conditions for which propulsion was observed.

6) Actually, what is the reason to choose the mucus solution in the study, why not use some model fluids that are well characterized and that show significant “rod-climbing” effects to clearly illustrate the propulsion mechanism?

Mucin was selected with the intention of demonstrating this effect within biological fluids, since they are ubiquitous throughout the human body and would need to be explored for any medical applications; but this was also the first fluid we noticed this phenomena to occur. It has the benefit of being able to demonstrate that certain medical treatments (drug delivery, hyperthermia, etc.) can be improved upon using our technique. We had intended to leave it to future researchers to examine the effects of other fluids. However, due to the referees’ suggestions to experimentally demonstrate the generality of the phenomenon, we have now investigated a low concentration polyacrylamide solution and found that microspheres could propel in a comparable manner. These results are similar to those in mucin, and are now presented in tandem with the original experiments throughout the revised manuscript. To summarize the new results, 1) microparticles in 0.25% polyacrylamide demonstrated an almost linear relationship with rotational frequency, 2) microparticles could be guided under feedback control within the fluid, 3) microparticles displayed similar behavior under a static field sweep when compared to those in synthetic mucus, 4) hysteretic behavior demonstrating two symmetry-breaking propulsive states was found, although only for higher rotational frequencies than in mucus, and 5) their velocity flow fields measured by PIV exhibited inward radial flow components consistent with rod-climbing-like effects. While there are differences between the fluids, these results do suggest that the symmetry breaking propulsion is a general phenomenon in nonlinearly viscoelastic fluids.

7) The propulsion velocities of different sized particles cannot be rationalized and need more experiments and explanations. The velocity clearly increases with sizes above 4-micron-diameter (Fig. 1(c)), it is contradictory with the size-dependent propulsion in porous media (what the authors wrote ‘smaller microparticles’ ability to move through the voids commonly found in mucus structures” in Page 5 Last Line). Particles at larger size than the pore size should not move (or move at lower speed), and the pore size seems to be around 4-5 microns according to SEM (Fig. 5(a)).

We note that at the concentration tested, mucin solution is a fluid, not a gel – therefore particles larger than the pore size can indeed move. At the referees’s request, we have re-

examined the size dependence for both 2 μm and 4 μm diameter microparticles in 4% mucin. A total of 12 and 14 individual microparticles were examined (with at least three trials each) including the original data sets for 2 μm and 4 μm diameter beads, respectively. In SI section 9, we describe these results:

We tested microparticles of several different diameters (2, 4, 8, and 10 μm) to determine if propulsion velocity was influenced by microparticle size [Fig. S16 (a-b)]; all data sets were averaged over at least six microparticles per diameter, per fluid, with at least 3 trials each. In 4% mucin as microparticle diameter increased, so did propulsion velocity, except that both 2 μm diameter microparticles and 4 μm diameter microparticles were nearly identical and had overlapping velocity curves. In 0.25% polyacrylamide the influence of microparticle diameter was less obvious, with 2 μm microparticles being faster than 4 μm microparticles, and 8 μm microparticles having a markedly non-linear relationship to frequency. Other than the 8 μm microparticles in polyacrylamide, for frequencies less than 10 Hz, the velocities of the differently sized microparticles in both fluids were all within a standard error of each other; but for frequencies above 10 Hz the velocity profiles of each diameter microparticle become more distinct from each other.

Based on these new observations, we agree with the referee's opinion that the quantitative aspects of size-dependence of the propulsion are not well-understood. However, as we expressed in our response to point 4, we regard quantitative understanding of the variability as out of the scope of our manuscript, which focuses on the surprising qualitative existence of propulsion and the physical mechanism that allows such symmetry breaking propulsion. We emphasize that particles of all sizes did exhibit propulsive behavior, and the propulsion could be controllably manipulated such as by 2D closed loop control [10 μm diameter, Fig. 2 (c-g)] or 3D open loop control [10 μm diameter in Fig. 2 (h) and 2 μm diameter in Fig. S10, in both mucin and polyacrylamide, respectively].

Therefore we have removed our original discussion about the possible explanations for size-dependence for the main text, where we now emphasize the robust existence of the phenomenon we wish to explain in this manuscript:

p 5: Propulsion was ubiquitous among rotated microparticles; it was observed for beads of several different diameters (2, 4, 8 μm) in mucin solutions of various concentrations (2%, 3%, 4%) and the polyacrylamide solution (see SI Section 9). Propulsion was observed for >90% of beads in mucin solution, and >60% of beads in polyacrylamide solution.

Should not the “transverse velocity” increase with the rotational frequency (Fig. S1(a) purple symbols), if the particle is touching the bottom as a rolling “wheel” (Page 4, third to the last line)?

We agree with the expectation discussed in this comment. In Fig S1(a), the microparticle is > 100 microns from the boundary and there is little transverse rolling motion. To directly show that the referee's expectation is correct and support the interpretation of the transverse motion as “rolling”, we tested particles close to the boundary (~1 micron away), leading to the results in Fig S2, which clearly demonstrate that the transverse velocity increases as rotational frequency increases.

8) The magnetic gradient of the coil system used in the study must be carefully measured in all directions and clearly reported. Even Helmholtz coil set-ups have inhomogeneities in the center of the field. Control experiments should be performed to prove that the propulsion has nothing to do with any magnetic gradient.

We agree with the referee that it is important to clearly rule out the alternative hypothesis that propulsion is caused by magnetic field gradients. The fact that propulsion diminished as frequency went to zero (Fig. 1, Fig. S1) implies that magnetic field gradients of the static field alone do not cause propulsion. We have taken the opportunity to explicitly state this in the main text on p 5, where we also refer to the additional control experiments described below which are now included in the SI:

Note that the propulsion decreased to almost zero as the rotational frequency approached zero, implying that static magnetic field gradients are not responsible for propulsion; see SI section 8 for further experiments that rule out this possibility.

In addition, we have now specified our experimental setup in detail in SI section 7, and measured the magnetic field within the working region. Plots of the 3D spatial variation of the magnetic field are displayed in Fig. S13, where the heat maps show the variation of the static field in each plane, while the histograms show the combined measured static field values from all three planes. In general, within the working region the magnetic field varies the most in the Z-direction, having a mean static field of 2.43 mT and a standard deviation of 0.34 mT when accounting for all Z-planes examined. In most of the presented propulsion experiments the propulsion is within the X-Y plane, where the X-coils and Y-coils had mean static field values of 2.16 mT and 2.08 mT, with standard deviations of 0.02 mT and 0.12 mT, respectively, when accounting for all Z-planes examined (see SI Section 7). These X-Y plane variations correspond to magnetic field gradients of at most 0.05 mT/m, which for a 10 μm diameter bead with a magnetic moment of $[-1.899 \times 10^{-12}, 6.277 \times 10^{-13}] \text{ A} \cdot \text{m}^2$ would lead to an estimated translational velocity of 0.0046 $\mu\text{m/s}$, orders of magnitude smaller than the propulsion velocities observed which are of order 1 $\mu\text{m/s}$. It would take 36 minutes for a 10 μm particle to travel the length of its diameter under these magnetic field gradients (see SI Section 8).

To further establish that magnetic field gradients were not responsible for the observed propulsion, we performed control experiments and compared the mean square displacement (MSD) of particles under an applied static field of 2 mT with and without an additional rotational magnetic field. The results are reported in SI Section 8. Two important points are evident: first, the MSD without rotation is three orders of magnitude smaller than the MSD with a rotating magnetic field, consistent with our previous observations of there being no propulsion without field rotation. Second, although with rotation the MSD scales ballistically (quadratically) with time, without the rotation the MSD scales diffusively (linearly) with time. We conclude based on this information that static field gradients are not responsible for the observed propulsion.

9) Why was a 4% mucus solution selected for the experiment? What are the propulsion results for other concentrations?

Human mucus can range in mucin glycoprotein concentration anywhere between 2-5%; we had just chosen 4% as a reasonable baseline value. A more detailed analysis showing microparticle velocity for 10 μm particles vs. frequency for two other concentrations of mucin

(2% and 3%) is now included in SI Section 9 and compared against the originally presented 4% mucin results. Symmetry breaking propulsion was present in all concentrations of mucin examined. Based on Fig. S15, it would appear that increasing mucin concentration positively affects overall velocity, while lower mucin concentrations can create nonlinear propulsion relationships with frequency. The exact limits of this behavior were not explored at this time.

10) Why do ferromagnetic particles need to be placed on a permanent magnet for magnetization (Page 16 Line 6)? Does it result in the agglomeration of the ferromagnetic particles?

As is typical for ferromagnetic materials, the particles are not necessarily magnetized initially (e.g., although iron is ferromagnetic, not all iron objects have a permanent magnetization). Application of a magnetic field larger than the coercive field (in our case, by placing them on a strong permanent magnet) consistently magnetizes the particles so they retain their remnant magnetization after being removed from the permanent magnet.

Agglomerations of these particles were rare since they were always introduced into the mucus medium at a very low concentration (although the likelihood of agglomeration increased with smaller microparticles since they were more concentrated). Since we rarely observed agglomeration, all experiments were conducted such that microparticles were far enough away from each other to avoid significant magnetic interactions between particles.

Reviewer #3 (Remarks to the Author, numbers added for convenience):

This manuscript describes a counter-intuitive form of propulsion, whereby the fore-aft symmetry of a rotating sphere is broken dynamically by the elasticity of the fluid medium. Viscoelastic media are important in biomedical contexts, especially, but propulsion in these media is studied much less than in Newtonian fluids, due in part to the difficulty of modeling their mechanics. The form of propulsion in the present experiments is, to my knowledge, unlike all others in microorganisms or considered for micro-robotics, wherein the propeller geometry and/or the driving select a direction; it appears to be a first for low-Reynolds-number propulsion. In these experiments the propulsive force is due to the nonlinear elastic hoop stresses generated around the rotating sphere, a mechanism that has been demonstrated elsewhere. What is new is that this force has a net direction not because of geometry, but because of the sphere's ongoing translation. The authors use the ubiquitous physics paradigm of spontaneous symmetry breaking, and show, fittingly, that a static magnetic field breaks the symmetry deterministically and selects the direction of motion. The supplemental information is thorough and useful. While it's unclear whether this form of propulsion is efficient and responsive enough to find practical applications in micro-robotics, drug delivery, etc., these experiments isolate an effect that is likely at play in more common and complicated forms of artificial and biological propulsion in viscoelastic fluids. This work has the potential to be of great interest to the broader propulsion and complex fluids communities.

1) My foremost concern has to do with justifying the choice of fluids in these experiments. The use of artificial mucus as the medium seems to needlessly complicate the experiments and detract from the proposed explanation of this newly-observed phenomenon, because these fluids have some structure on the scale of the spheres and their trajectories, and have unusual interactions related to their biological function. Mucus seems more appropriate for a proof-of-concept for applications, rather than a systematic study to elucidate the mechanism. Why did the authors choose mucus, and would these experiments have also worked with polymer that is not so specialized, such as polyacrylamide?

Mucin was selected with the intention of demonstrating this effect within biological fluids, since they are ubiquitous throughout the human body and would need to be explored for any medical applications; but this was also the first fluid we noticed this phenomena to occur. It has the benefit of being able to demonstrate that certain medical treatments (drug delivery, hyperthermia, etc.) can be improved upon using our technique. We had intended to leave it to future researchers to examine the effects of other fluids. However, due to the referees' suggestions to experimentally demonstrate the generality of the phenomenon, we have now investigated a low concentration polyacrylamide solution and found that microspheres could propel in a comparable manner. These results are similar to those in mucin, and are now presented in tandem with the original experiments throughout the revised manuscript. To summarize the new results, 1) microparticles in 0.25% polyacrylamide demonstrated an almost linear relationship with rotational frequency, 2) microparticles could be guided under feedback control within the fluid, 3) microparticles displayed similar behavior under a static field sweep when compared to those in synthetic mucus, 4) hysteretic behavior demonstrating two symmetry-breaking propulsive states was found, although only for higher rotational frequencies than in mucus, and 5) their velocity flow fields measured by PIV exhibited inward radial flow components consistent with rod-climbing-like effects. While there are differences between the

fluids, these results do suggest that the symmetry breaking propulsion is a general phenomenon in nonlinearly viscoelastic fluids.

2) Similarly, the authors seem to be very reluctant to identify nonlinear fluid elasticity as the key ingredient that makes this form of propulsion possible—it is not mentioned in the abstract or introduction, even at the mention of [elastic] hoop stresses. This is confusing and makes one wonder whether shear-thinning or any of the special additional properties of mucus (related to its biological function) are important or sufficient for propulsion. I think that this choice of emphasis does a disservice to the modeling component of the paper, and again detracts from this work's impact.

We did not mean to deemphasize nonlinearity; indeed since hoop stresses are a result of the nonlinear 1st normal stress difference, and rod-climbing depends on the nonlinear 1st and 2nd normal stress differences, we thought that it was clear that fluid nonlinearity was the key ingredient necessary for this type of propulsion, especially since (as the referee points out) the model requires nonlinearity of the constitutive law. We thank the referee for suggesting that it may be helpful to explicitly state the importance of nonlinearity, which we now do in the abstract:

We propose and experimentally corroborate a propulsion mechanism for these spherical microparticles, the simplest microswimmers to date, arising from nonlinear viscoelastic effects in rotating flows similar to the rod-climbing effect.

In the introduction on p. 3 we have added the following description:

We propose a physical mechanism for symmetry breaking that arises from nonlinear viscoelastic effects in rotating flows, similar to the rod-climbing effect, which pushes the sphere along its rotation axis.

We have also significantly clarified our discussion of the mechanism and importance of the nonlinear viscoelastic effects on p 11:

we propose that the symmetry breaking mechanism is caused by nonlinear viscoelastic stresses that arise in rotating flows, similar to those responsible for the “rod-climbing” effect. Around a vertical rotating rod in a polymeric fluid, nonlinear first normal stress differences cause excess circumferential hoop stresses along the circular streamlines, while second normal stress differences cause excess radial stress; both can squeeze the fluid upwards around the rod²¹. We note that both polyacrylamide and mucin solutions display first and second normal stress differences that lead to rod-climbing-like effects (see Methods, Fig. 5 and SI Section 10). For a rotating and translating sphere, fluid material also advects past the sphere, so that fluid elements at the back of the sphere have spent more time circling the sphere than those at the front; thus, the back of the sphere has larger nonlinear viscoelastic stresses than the front, producing a net squeezing force that propels the sphere forward. A force-free symmetry-breaking translational state occurs when this propulsive force balances the drag force from translation.

3) Finally, it was unclear why the authors did some experiments with a methylcellulose solution, or how the results support their conclusions. Why was it important to have a second Newtonian system—especially a complex fluid that is only approximately Newtonian?

This was included as an additional Newtonian fluid control which was polymeric.¹⁹ Dilute methylcellulose solutions have Newtonian behavior at low shear rates, which extend to higher shear rates as methylcellulose concentration decreases. According to Moreira et al (DOI: <https://doi.org/10.1021/acsmacrolett.5b00150>), for a 0.75% methylcellulose (MW 360,000) solution the Newtonian behavior extends up to shear rates of 100 s^{-1} . Thus our methylcellulose solution (0.2%, MW 86,000) can be expected to behave as a Newtonian fluid throughout the range of our experiments (shear rates up to 120 s^{-1} , see response to 10).

I have a handful of more substantive concerns and suggestions that the authors should consider in their revisions, as well as several more minor issues.

4) The authors are unfortunate in that ferromagnetism has some of the same behaviors as they observe for propulsion, and this will probably occur to many readers. Specifically, the text needs to identify why these particles' magnetic hysteresis cannot be the origin of the hysteresis in Fig. 3d. (Variability in magnetic permanence is, at least superficially, an alternate explanation for Fig. 3c.)

We thank the referee for pointing out this possible alternative explanation, which should be addressed in the paper. Varying magnetization does not explain the symmetry-breaking propulsion for the following reasons.

a) While magnetic materials can display hysteretic magnetization upon application of a reversing static magnetic field, the situation here is quite different. In our hysteresis sweeps, only the static component is varied between -2 mT and 2 mT, while there is always a rotating magnetic field present of magnitude $\sim 11.68 \text{ mT}$ or $\sim 10.89 \text{ mT}$ in the 15 Hz (mucin) or 40 Hz (polyacrylamide) experiments, respectively. It would be inconsistent for a variation of 2 mT in static field to be enough to change the magnetization (i.e., be close to the coercive field strength) while the total magnetic field is changing by much more than that over time due to the rotational component. Indeed, to be quantitative, we magnetize our ferromagnetic beads using a permanent magnet, and afterwards their magnetization is unlikely to change for this range of magnetic field magnitudes used in the experiment (3.11 – 15.87 mT), which are far below the coercive field strength (68 mT (Hiller, DOI: <https://doi.org/10.1063/1.324824>), which is consistent with our experience magnetizing and demagnetizing beads).

We have added to the main text the following statement explaining the above (p 11).

The hysteretic behavior of propulsive velocity is reminiscent of those seen in other spontaneously symmetry breaking phenomena, such as ferromagnetism, but ferromagnetic behavior cannot explain our observations, since the coercive fields of 68 mT²⁵ required to magnetize and demagnetize our beads are much larger than the magnitudes of combined rotational and static fields used in our experiments (3.11 – 15.87 mT).

b) We point out that even if the magnetization of the spherical bead changed like a hysteretic ferromagnet during our hysteresis sweep, at zero static field there would still be some nonzero moment for both the upwards and downwards sweep. Because of the spherical symmetry it does not matter what direction the moment points in; the opposite propulsion observed at zero static field would still constitute a pair of symmetry broken propulsive states.

5) The supplemental information details a model based on a general framework by Giesekus. This earlier work shows the possibility of a propulsive force and spontaneous symmetry-breaking, albeit for an idealized fluid and for perturbative velocities. I think that this is worth mentioning in the introduction of the main text. It will hardly detract from the novelty of the present work, and it will increase its significance by making readers more confident that this possibility arises from generic nonlinearities of viscoelastic fluids but was hiding in plain sight.

We have now added to the last paragraph of the introduction a reference and citation to Giesekus' work (Ref 14) as suggested:

The mechanism is corroborated by comparison to existing theoretical analyses of rotating and translating spheres in generic third-order fluids¹⁴.

6) The model appears to show plainly that a rotating and non-translating sphere ($\Omega > 0$, $U = 0$) can be linearly unstable to translation. If true, this is worth pointing out in the main text, since it directly leads to spontaneous symmetry-breaking, and is the crux of a dynamical analysis of the system (see e.g. Strogatz, "Nonlinear Dynamics and Chaos").

We agree with the referee, and have added to p. 12:

If $C > 0$, then there is a nonlinear thrust in the same direction as translation which could stabilize a translating symmetry-broken state, i.e, could make a rotating non-translating sphere unstable to translation; while if $C < 0$, then nonlinear effects stabilize the zero-velocity state.

7) Can the authors rule out magnetic interactions between spheres in their experiments? Each micrograph shows only one sphere, but it's unclear how that was achieved.

The microparticles were far enough away from each other such that magnetic and fluid interactions between particles were minimized; it was very rare for us to observe agglomerations of microparticles.

8) It seems that the experimental protocols implicitly rules out the important possibility that magnetic field gradients are causing the appearance of propulsion. The authors should consider making this explicit.

We agree with the referee that it is important to clearly rule out the alternative hypothesis that propulsion is caused by magnetic field gradients. As recognized by the referee, the fact that propulsion diminished as rotational frequency went to zero (Fig. 1, Fig. S1) implies that magnetic field gradients alone due to the static field do not cause propulsion. We have taken

the suggestion to explicitly state this in the main text on p 5 where we also refer to the additional control experiments described below which are now included in the SI:

Note that the propulsion decreased to almost zero as the rotational frequency approached zero, implying that static magnetic field gradients are not responsible for propulsion; see SI Section 8 for further experiments that rule out this possibility.

In addition, we have now specified our experimental setup in detail in SI section 7 and measured the magnetic field within the working region. Plots of the 3D spatial variation of the magnetic field are displayed in Fig S13, where the heat maps show the variation of the static field in each plane, while the histograms show the combined measured static field values from all three planes. In general, within the working region the magnetic varies the most in the Z-direction, having a mean static field of 2.43 mT and a standard deviation of 0.34 mT when accounting for all Z-planes examined. In most of the presented propulsion experiments the propulsion is within the X-Y plane, where the X-coils and Y-coils had mean static field values of 2.16 mT and 2.08 mT, with standard deviations of 0.02 mT and 0.11 mT, respectively, when accounting for all Z-planes examined(see SI Section 7). These X-Y plane variations correspond to magnetic field gradients of at most 0.05 mT/m, which for a 10 μm diameter bead with a magnetic moment of $[-1.899 \times 10^{-12}, 6.277 \times 10^{-13}] \text{ A} \cdot \text{m}^2$ would lead to an estimated translational velocity of 0.0046 $\mu\text{m/s}$, orders of magnitude smaller than the propulsion velocities observed which are of order 1 $\mu\text{m/s}$. It would take 36 minutes for a 10 μm particle to travel the length of its diameter under these magnetic field gradients (see SI Section 8).

To further establish that magnetic field gradients were not responsible for the observed propulsion, we performed control experiments and compared the mean square displacement (MSD) of particles under an applied static field of 2 mT with and without an additional rotational magnetic field. The results are reported in SI Section 8. Two important points are evident: first, the MSD without rotation is three orders of magnitude smaller than the MSD with a rotating magnetic field, consistent with our previous observations of there being no propulsion without field rotation. Second, although with rotation the MSD scales ballistically (quadratically) with time, without the rotation the MSD scales diffusively (linearly) with time. We conclude based on this information that static field gradients are not responsible for the observed propulsion.

9) Additionally, it's unclear why the motion of a sphere without actuation that the authors observe is attributed to convective flow and not field gradients.

The reason that we attribute the motion of spheres without rotation to flow rather than gradients is that such motion was not consistently observed. While any gradients in the magnetic field should be consistent from experiment to experiment, flows in our sample chamber (although we attempted to minimize them) are harder to control and can arise from many sources included density variations from optical heating, evaporation, sedimentation, etc.

Inspired by this comment, we performed additional control experiments in which we tried to minimize the convective flows. We found that by better matching the bead densities and preventing sedimentation using a 15% NaCl solution rather than 30% solution, sedimentation and uncontrolled 3D flows could be much reduced. In the revised manuscript, we now use these four new 15% NaCl control data sets (each with at least three trials each) in Fig 1. The fact that we could remove much of the motion at zero frequency by controlling sedimentation supports our attribution of the motion without actuation to convective flow.

We also performed additional control experiments in 0.2% methylcellulose in which we observed much lower convective flows. In the data shown in Fig 1 of the revised manuscript, we now include these additional experiments; we selected the data to include from both our new and old control experiments by removing individual trials that had velocities greater than 0.25 $\mu\text{m/s}$ as the frequency approached zero. This criterion removed 6 of the 35 trials over 10 beads now presented in the paper; all six of the removed trials were from the originally presented data.

In the caption to Fig 1, we have now also included coefficient of determination values (r^2) that indicate the control experiments have velocities which do not correlate with frequency, which supports the conclusions that they do not exhibit propulsion:

Fig. 1. Propulsion characteristics of microparticles. (a) Reflection about the symmetry plane leaves the geometry and rotation unchanged but reverses the propulsion velocity U to U' . If there is only one state, $U = U' = 0$, but spontaneous symmetry breaking results in two propulsive states with equal and opposite velocities U and U' . Propulsion velocity vs. rotational magnetic field frequency for 10 μm diameter microparticles in (b) 4% mucin and (c) 0.25% polyacrylamide. Control experiments using 10 μm diameter microparticles were performed in two Newtonian fluids: 15% NaCl and 0.2% methylcellulose; both were plotted in both (b) and (c) to compare with the propelling microparticles. The coefficient of determination (r^2) values for the linear fit in 4% mucin solution and 0.25% polyacrylamide solution were 0.9650 and 0.9449, respectively, indicating a strong linear correlation of velocity and frequency. In contrast, the r^2 values for the linear fits to controls in 15% NaCl and 0.2% Methylcellulose solution are 0.3773 and 0.2036, respectively, indicating little correlation with frequency. Error bars represent standard error.

Further confirmation that low frequency motion was not the result of static magnetic fields can be found in SI Sections 7 and 8 where we measure the gradient of the applied static field to be 0.004 $\mu\text{m/s}$ and the mean square displacement analysis for non-rotating particles in mucus were approximately diffusive [see response to 8) above].

10) I may have missed it, but it seems like there is very little effort to connect the measured rheology of these fluids with strain rates in the experiments. Could the authors comment on the shear strain rate regime in which these experiments take place?

For a rotating microsphere in the low Reynolds number limit, the Newtonian solution has a maximum strain rate near the sphere surface which is equal to the angular velocity of the sphere. Therefore, the frequency of rotation is directly related to the strain rate during our experiments. So the shear rate is between 1 to 120 s^{-1} meaning we are in the shear thinning regime of Fig. (5). Accordingly we have added to the Methods (p. 19) section the following text:

Note that the strain rate expected nearby a rotating sphere is equal to the angular velocity of the sphere; for our experiments this varies from 0-120 s^{-1} .

Minor issues

- In Figs. 1, 2, 4, and S2, the authors should strongly consider replacing the “jet” colormap with a perceptually uniform one, which is available in all major scientific plotting packages. Perceptually uniform colormaps have been the best practice for roughly the past decade, since

they are easier to interpret quantitatively, and as a bonus, do not needlessly hinder readers who are colorblind or who have black-and-white printers.

We replaced 'jet' colormap with a 'copper' color map. This should alleviate most issues.

- Paragraph beginning at bottom of p. 7: In this paragraph I was confused about the direction of the static magnetic field and how it related to the propulsion direction. It's also unclear why "(mT)" is included in the first sentence of the paragraph.

We corrected our word choice in the first sentence of the paragraph (now the 1st paragraph on p 6) to clarify that:

application of a magnetic field controlled propulsion to occur in the direction of the static magnetic field, so we next investigated the dependence of propulsion on the static magnetic field.

- Caption of Fig. 3(a): Range should be -5 to 5 mT.

This was fixed.

- p. 9: At the beginning of this paragraph, in "To understand how a static field might be able to control the propulsion direction", "understand" seems like a poor word choice since this remains a major open question.

This word was replaced by "elaborate on." (p. 8)

- Top of p. 11: Should be "Stokes' Law" or "Stokes's Law". Or one could write "Stokes drag on a translating sphere."

This was corrected.

- In Fig. 4, why not rotate (a) or (b) so that they have the same propulsion direction? The figure would then better support the comparison in the text.

This suggestion has been implemented.

- It's appropriate in Fig. 4 to scale velocities by the maximum velocity in each experiment, as the authors do. However, this leaves the reader without a sense of how these velocities compare with the spheres' translation velocities that are the primary concern of the paper. Can this information be included?

This information has been added to SI Section 13, which includes the maximum velocity for each μ PIV figure in the main manuscript and the additional μ PIV figures added to the SI.

- Can the authors comment further on the apparent super-linear increase in propulsion velocity with sphere size? What would limit the propulsion of a mm-scale sphere, or larger?

At this time we do not have a detailed understanding of how the particle size affects the propulsion velocity. We emphasize that the focus of this paper is on the robust and surprising existence of symmetry breaking propulsion; we do not try to completely understand all the effects which could influence the propulsion velocity. (Also see response to Referee 2, 4th and 7th comments.)

We have revised the discussion (p. 15) to explicitly mention that the size dependence is worth future investigation:

In this manuscript we focused on the existence and mechanism for symmetry breaking propulsion; further work is needed to completely understand the conditions (geometry, rotation frequency, fluid properties) required for symmetry breaking to occur, the size dependence and effect of boundaries of propulsion velocities, as well as how fluid properties and actuation history control the hysteresis and selection of propulsion direction.

Inertial effects typically oppose rod-climbing effects, so they set an upper size limit for this propulsion mechanism. However, we do not know if there may be other physical limits that come into play.

- On p. 16, at “the ferromagnetic particle solution,” “suspension” would be a much better term.

Done.

- In the “Contributions” section (p. 20), “principle investigator” should be “principal investigator” (2 instances).

Corrected.

REVIEWERS' COMMENTS

Reviewer #1 (Remarks to the Author):

A carefully prepared manuscript that answers all the points raised in the reviewers comments.

The authors have gone to considerable extent and effort to answer the points raised in the initial review. Whilst the comment about the need for simulations, i agree with the author that is a study in itself.

I believe this is now suitable for publication.

Reviewer #2 (Remarks to the Author):

The authors have extended their work and added additional experiments to show more clearly that the effect they observe is robust. The changes have significantly improved the manuscript. The observed translation of a rotating sphere in a non-Newtonian fluid in the presence of a rotating and static magnetic field is an interesting and surprising phenomenon. I agree with the authors that a full simulation is not feasible. I do not fully agree that their experiment is really far from walls. I tried very hard to understand what exactly happens, in particular, what causes the initial translation or symmetry-breaking. The authors go on to show that when there is some fore-aft symmetry-breaking that the flows and forces will support translation, but what happens right at the start of the experiment when the sphere moves in one direction? Many questions remain and the 'explanations' the authors offer are still mainly speculation. I think more experiments are needed to properly figure out what the real reason for the translation is. However, the effect is interesting and may stimulate others to consider the phenomenon.

Reviewer #3 (Remarks to the Author):

I thank the authors for their careful attention to the referees' concerns. All of my concerns have been adequately addressed in this revision, most notably by the crucial addition of polyacrylamide experiments. The evidence presented is now appropriate to the authors' claims about the fundamental mechanism for this propulsion, and ensures that this paper will have a significant impact on the broader propulsion and complex fluids communities, as detailed in my initial review. I am happy to recommend publication in Nature Communications.

Nathan C. Keim

Reviewers' comments:

Reviewer #1 (Remarks to the Author):

A carefully prepared manuscript that answers all the points raised in the reviewers comments.

The authors have gone to considerable extent and effort to answer the points raised in the initial review. Whilst the comment about the need for simulations, i agree with the author that is a study in itself.

I believe this is now suitable for publication.

We thank the reviewer for their kind remarks and appreciate their time and effort during these difficult times.

Reviewer #2 (Remarks to the Author):

The authors have extended their work and added additional experiments to show more clearly that the effect they observe is robust. The changes have significantly improved the manuscript. The observed translation of a rotating sphere in a non-Newtonian fluid in the presence of a rotating and static magnetic field is an interesting and surprising phenomenon.

We thank the reviewer for pushing us to demonstrate this effect in other nonlinearly viscoelastic fluids and are grateful that its addition has improved the manuscript.

I agree with the authors that a full simulation is not feasible.

We thank the reviewer for understanding the magnitude of any proposed numerical simulations. We agree that they are an important target for future work on this system, and plan to collaborate with other groups in the future to better quantify this effect using numerical simulations.

I do not fully agree that their experiment is really far from walls.

While the microparticles investigated during these papers did sink due to gravity driven sedimentation with time, all microparticles were far from the walls ($> 100 \mu\text{m}$) during all experiments unless specified. In Section 1 of the SI we also demonstrate that boundary effects become small as the distance from the boundary increases, decaying at around 20-30 μm , and are certainly small at 100 μm (see Fig. S1).

I tried very hard to understand what exactly happens, in particular, what causes the initial translation or symmetry-breaking. The authors go on to show that when there is some fore-aft symmetry-breaking that the flows and forces will support translation, but what happens right at the start of the experiment when the sphere moves in one direction? Many questions remain and the 'explanations' the authors offer are still mainly speculation. I think more experiments are needed to properly figure out what the real reason for the translation is.

If, as we claim, the propulsion is due to a symmetry breaking that makes the non-propulsive state unstable, then a powerful consequence is that any perturbation is sufficient at the start of rotation to initiate dynamics that lead to one of the two symmetry broken propulsive states. Indeed, in other symmetry breaking systems it is very common that some small random perturbation eventually leads to the selection of one of the symmetry broken states. Therefore, while the referee is correct that we have not identified the initial perturbation (which could be different for each particle or instance of propulsion), we view it as of secondary concern, since the eventual propulsion phenomenon is independent of the initial triggering perturbation.

To emphasize this point, we now write on page 12 of the manuscript:

If $C > 0$, then there is a nonlinear thrust in the same direction as translation which could stabilize a translating symmetry-broken state, i.e., a rotating non-translating sphere could become unstable to translation; while if $C < 0$, then nonlinear effects stabilize the zero-velocity state. Due to the instability of the non-translating state, upon rotating an initially stationary sphere any small perturbation would lead to the symmetry-broken translating state.

(The first sentence was already there; the second sentence is an addition to the revised manuscript.)

However, the effect is interesting and may stimulate others to consider the phenomenon.

We appreciate the referee for diving deep into this research and challenging us; we hope that the referee is correct and that this research will stimulate the field at large in the coming years.

Reviewer #3 (Remarks to the Author):

I thank the authors for their careful attention to the referees' concerns. All of my concerns have been adequately addressed in this revision, most notably by the crucial addition of polyacrylamide experiments. The evidence presented is now appropriate to the authors' claims about the fundamental mechanism for this propulsion, and ensures that this paper will have a significant impact on the broader propulsion and complex fluids communities, as detailed in my initial review. I am happy to recommend publication in Nature Communications.

We thank the reviewer for their supportive comments and for taking the time to provide us with feedback that helped strengthen our initial manuscript. We sincerely hope that this work and its future derivations greatly contribute to fluid dynamics and microrobotics communities at large.